# Vsb1, Ypq1, and Ypq2 control dynamic cationic amino acid storage in the yeast vacuole

Evi Zaremba[1,2] , Fabienne Vierendeels[1], Raphaël Dutoit[1] , Elisabeth Bodo[1], Ersilia Bifulco[3] , Catherine Tricot[1], Elad Noor[4] , Bruno André[2], Evgeny Onischenko[3] , Melody Cools[1]

**Although the yeast vacuole plays a crucial role in storing and mobilizing cationic amino acids (CAA), CAA transport at the vacuolar membrane remains only partially characterized. Here, by combining analysis of CAA pools, uptake and permeabilization assays, we establish Vsb1 as the principal vacuolar lysine transporter, enabling its strong accumulation in the vacuole when mitigating its toxicity. We further show that, although Ypq1 can mediate proton-independent vacuolar lysine import, it mainly functions as a lysine exporter necessary for lysine mobilization under conditions of lysine scarcity and is down-regulated as lysine stores are exhausted. Using quantitative models based on dynamic metabolic labeling, we further show that, surprisingly, in growing cells, CAA rapidly exchange between vacuolar and cytosolic compartments, a process involving the export activity of Ypq1 and its paralogue Ypq2, specific for lysine and arginine, respectively. Together, our findings reveal the unexpectedly complex function of Vsb1 and Ypq1/2 as the key transporters mediating dynamic vacuolar CAA storage.**

## Introduction

Microorganisms inhabit dynamic environments and face numerous nutritional challenges. In nutrient-poor conditions, they must contend with intense competition for essential resources such as nitrogen, phosphorus, and trace metals (Watanabe et al, 2008; Broach, 2012; Sun et al, 2022). This requires highly efficient scavenging mechanisms (Kim & Klionsky, 2000), specific transport systems (Horák, 1997; Kschischo et al, 2016), and the capacity to store large quantities of metabolites (Klionsky, 1990) when nutrients are abundant and to recycle building blocks of macromolecules during periods of scarcity (Li & Vierstra, 2012; Reggiori & Klionsky, 2013). However, these processes require precise regulation when conditions shift to nutrient-rich environments, as excessive metabolite accumulation can result in toxicity (Li & Kane, 2009).

In budding yeast (*Saccharomyces cerevisiae*), the vacuole serves as a principal storage organelle for a variety of metabolites, including amino acids, polyphosphates, and metal ions (Klionsky, 1990), allowing cells to withstand long periods of nutritional deprivation (Li & Kane, 2009). Furthermore, because it can sequester harmful compounds, the vacuole significantly contributes to detoxification and resistance to toxic metabolites (Li & Kane, 2009; Jézégou et al, 2012; Cools et al, 2020). Functionally analogous to mammalian lysosomes, the yeast vacuole also hosts hydrolases that degrade macromolecules (Klionsky, 1990; Li & Kane, 2009) and sometimes entire organelles delivered via autophagy (Reggiori & Klionsky, 2013). Its dual role in storage and metabolite recycling has presumably driven the evolution of specialized vacuolar transport systems (Li & Kane, 2009).

The activity of most vacuolar importers is thought to depend on the proton gradient generated by the V-ATPase (Shimazu et al, 2005; Sekito et al, 2014b; Kawano-Kawada et al, 2019, 2021; Cools et al, 2020), a multiprotein complex that hydrolyzes ATP to ADP to pump protons into the vacuole. Its Vph1 subunit is essential for both assembly and activity (Nishi et al, 2003; Li & Kane, 2009). During steady-state growth in a rich nitrogen medium, vacuolar arginine import is primarily mediated by Vsb1, a member of the SLC26/SulP family of transporters (Cools et al, 2020). However, under nitrogen starvation, Vsb1 activity is inhibited and Ypq2 exports arginine from the vacuole to the cytosol, where it serves as an alternate nitrogen source, crucial for long-term survival (Cools et al, 2020). Ypq2 belongs to the PQ-loop family of transporters, which are characterized by a conserved repeated PQ motif critical to their activity (Jézégou et al, 2012).

Similarly to arginine, lysine is predominantly stored in the vacuole (Messenguy et al, 1980). Its uptake into the cell is mediated by the high-affinity plasma membrane permease Lyp1, as well as Can1 and Gap1 under specific conditions (Grenson, 1966; Grenson et al, 1966; Sychrova et al, 1993). Lysine biosynthesis is tightly regulated: expression of lysine biosynthetic genes is induced in response to lysine deprivation (Wolfner et al, 1975; Urrestarazu et al, 1985; Ramos et al, 1988) and deletion of any gene in this pathway renders cells lysine auxotrophs, dependent on external

---

[1]Labiris, Brussels, Belgium   [2]Molecular Physiology of the Cell, Université Libre de Bruxelles (ULB), Biopark, Gosselies, Belgium   [3]Department of Biological Sciences, University of Bergen, Bergen, Norway   [4]Department of Plant and Environmental Sciences, Weizmann Institute of Science, Rehovot, Israel

Correspondence: mcools@spfb.brussels

lysine for growth (Sinha & Bhattacharjee, 1971). Interestingly, unlike most amino acids, lysine cannot serve as a nitrogen source for yeast (Watson, 1976), although it can act as a precursor for high-level synthesis of polyamines (Olin-Sandoval et al, 2019). In addition, lysine can be toxic to yeast in certain contexts, such as growth on poor nitrogen sources, although the reasons for this toxicity remain unclear (Sumrada & Cooper, 1978; Cooper et al, 1979; Thomas & Ingledew, 1994).

Various published reports have identified vacuolar lysine transporters. These studies, particularly those exploiting uptake assays in isolated vacuoles derived from mutant strains, provided genetic evidence that Vsb1, the Major Facilitator Superfamily transporters Vba1, Vba2, and Vba3, as well as Ypq1, a transporter closely related to Ypq2, catalyze proton-dependent lysine import at the vacuolar membrane (Shimazu et al, 2005; Sekito et al, 2014b; Manabe et al, 2016; Kawano-Kawada et al, 2019, 2021). These were complemented with total and vacuolar pool measurements which confirmed that Vsb1 is essential for vacuolar lysine accumulation and that a conserved residue of its first transmembrane (TM) α-helix, Asp-223, is essential for its transport activity (Kawano-Kawada et al, 2021). In addition, $^{14}$C-lysine transport was detected in liposomes reconstituted with purified Ypq1 (Arines et al, 2024). Far less is known about lysine export from the vacuole but total pool measurements have suggested a role of Avt4 in lysine export under nitrogen starvation (Sekito et al, 2014a). The fact that the *YPQ3* gene, encoding another member of the PQ-loop family, is induced in response to lysine withdrawal has marked it as another putative lysine exporter (Jézégou et al, 2012). Thus, despite the identification of several proteins capable of mediating lysine transport across the vacuolar membrane, the primary lysine importer and exporter remain unclear. Moreover, how vacuolar CAA transporters work together to maintain cellular CAA homeostasis and balance cytosolic and vacuolar intracellular CAA pools under different physiological conditions is not well understood.

In this study we combine structural modeling coupled with mutagenesis experiments, biochemical transport assays, quantitative pool measurements and dynamic labeling to explore the dynamics of vacuolar CAA transport during active growth and under stress conditions. We confirm that, under state conditions, in addition to its established role in arginine transport, Vsb1 acts as the main proton-gradient dependent importer responsible for vacuolar accumulation of lysine. Its RmlC-like fold domain and key residues Tyr-227 and Glu-278 (in addition to Asp-223) within the transmembrane domain were found to be essential for its activity. Furthermore, we show that when Vsb1 accumulates lysine in the vacuole to shield cells from the toxic effects of its excess, in contrast to what was previously reported, Ypq1 mediates vacuolar lysine export and is essential for mobilizing vacuolar lysine stockpile during its scarcity. Finally, by implementing quantitative models based on dynamic labeling, a methodological approach not previously used to examine vacuolar transport in live cells, we show that, under active growth conditions, Ypq1 and Ypq2 enable surprisingly high exchange rates of lysine and arginine, respectively, corresponding to approximatively one total pool of cellular CAA per hour.

# Results

## Vsb1 and Ypq1 are involved in lysine transport across the vacuolar membrane

We and others have previously demonstrated that arginine transport at the vacuolar membrane is catalyzed by the putative proton antiporter Vsb1 and by the Ypq2 exporter (Cools et al, 2020; Kawano-Kawada et al, 2021). However, the mechanisms of vacuolar transport of lysine, another major CAA, remain only partially characterized. As most candidate vacuolar lysine transporters have been proposed to function as proton antiporters (Sato et al, 1984; Sekito et al, 2014b; Cools et al, 2020), we investigated whether lysine accumulation in the vacuole requires the vacuolar proton gradient (Sato et al, 1984). Thus, we analyzed the effect of the *VPH1* deletion on the total pool of soluble lysine (Preston et al, 1989; Manolson et al, 1992; Li & Kane, 2009). As shown by previous studies, this analysis reliably reflects vacuolar lysine content, as at least 80–90% of free cellular lysine resides in the vacuole (Messenguy et al, 1980; Cools et al, 2020). UPLC analysis of soluble metabolites revealed that deletion of *VPH1* resulted in a significant reduction in soluble lysine content (Fig 1A), implying that lysine accumulation requires the proton gradient and that vacuolar lysine importers are likely H$^+$-antiporters.

We next sought to investigate the role of eight putative lysine transporters in vacuolar lysine accumulation: Avt4 (Sato et al, 1984; Russnak et al, 2001; Yang et al, 2006; Sekito et al, 2014a), Vba1-2-3 (Ohsumi & Anraku, 1981; Sato et al, 1984; Shimazu et al, 2005), Vsb1 (Cools et al, 2020; Kawano-Kawada et al, 2021) and Ypq1-2-3 (Jézégou et al, 2012; Sekito et al, 2014b; Manabe et al, 2016; Arines et al, 2023 *Preprint*). Among them, Avt4 was discarded because our reference strain (Σ1278b) carries an insertion that leads to a premature stop codon in the *AVT4* gene (Cherry et al, 2012). We, therefore, focused on the remaining seven proteins and assessed their contribution to lysine accumulation by measuring the total soluble lysine pool in the corresponding deletion mutants (Fig 1A). Among the tested mutants, the *vsb1Δ* strain displayed the strongest reduction in the intracellular lysine pool, pointing to a defect in the vacuolar accumulation of lysine. Interestingly, the *ypq1Δ* and triple *ypq1-2-3Δ* mutants showed the opposite effect, namely an over-accumulation of intracellular lysine, which may point to an impaired lysine mobilization from the vacuole. The other analyzed deletion mutants (*ypq2Δ, ypq3Δ, vba1-2-3Δ*) displayed close to no impact on lysine accumulation, implying that Vsb1 and Ypq1 are likely the key players in lysine transport across the vacuolar membrane.

To further assess the role of Vsb1 and Ypq1 in the establishment of cellular lysine levels, total soluble lysine pool measurements were, in addition, performed in strains transformed with a plasmid expressing *VSB1* or *YPQ1* under the strong constitutive *TDH3* promoter (McAlister & Holland, 1985). *VSB1* overexpression led to a threefold increase in the total lysine pool, whereas *YPQ1* overexpression reduced it to levels comparable witth that of the *vsb1Δ* strain (Fig 1A). These findings further support the conclusion that Vsb1 and Ypq1 play important roles in vacuolar lysine transport. Incidentally, neither deletion nor overexpression of *VSB1* or *YPQ1*

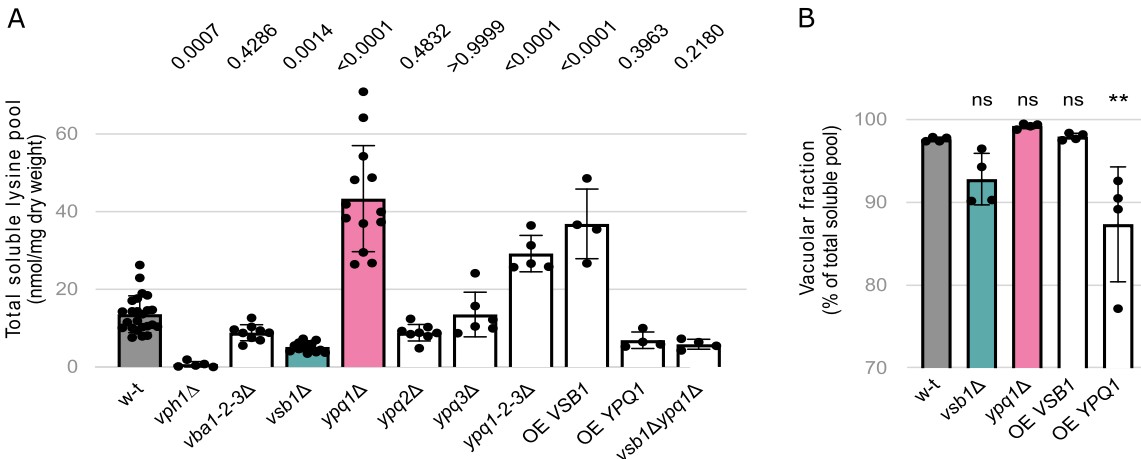

**Figure 1. Vsb1 and Ypq1 are involved in lysine transport across the vacuolar membrane.**
**(A)** The intracellular lysine content was measured in the WT, *vph1Δ*, *vba1-2-3Δ*, *vsb1Δ*, *ypq1Δ*, *ypq2Δ*, *ypq3Δ*, *ypq1-2-3Δ*, OE *VSB1* (*vsb1Δ* complemented with pFV438 plasmid, allowing *VSB1* overexpression under *TDH3* promoter), OE *YPQ1* (*ypq1Δ* complemented with pFV445 plasmid, allowing *YPQ1* overexpression under *TDH3* promoter) and *vsb1Δypq1Δ* strains. The *P*-values were calculated by a one-way ANOVA with post hoc comparison tests with the WT strain (n = 4–24 biological replicates). **(B)** Percentage of the total lysine pool in the vacuolar fraction. The cytosolic and vacuolar fractions were separated using the copper chloride method. Amino acids were measured in both fractions in the WT (WT complemented with pFL38, an empty plasmid), *vsb1Δ* (*vsb1Δ* complemented with pFL38 plasmid, an empty plasmid), *ypq1Δ* (*ypq1Δ* complemented with pFL38 plasmid, an empty plasmid), OE *VSB1* (*vsb1Δ* complemented with pFV438 plasmid, allowing *VSB1* overexpression under *TDH3* promoter) and OE *YPQ1* (*ypq1Δ* complemented with pFV445 plasmid, allowing *YPQ1* overexpression under *TDH3* promoter) strains. Results are normalized using glutamate quantities (ns: $P > 0.05$; **: $P < 0.0021$ by one-way ANOVA with post hoc comparison tests with the WT strain) (n = 4 biological replicates).

affect cell growth as doubling time remained the same across strains and conditions (Table S1).

Because our conclusions rely on total soluble lysine measurements assuming that it is mainly vacuolar (Kitamoto et al, 1988), we next examined these effects using direct measurements of the vacuolar content of lysine. Vacuolar and cytosolic fractions of the total lysine pools of the *vsb1Δ* and *ypq1Δ* mutants and their overexpression strain counterparts were extracted by the $Cu^{2+}$ permeabilization method and quantified by UPLC. Here, glutamate served as a fiducial cytosolic marker to control for permeabilization efficiency (Messenguy et al, 1980). More than 85% of lysine was recovered in the vacuolar fraction across all strains, validating the use of the total pool as a proxy for vacuolar pool (Fig 1B). Interestingly, both *VSB1* deletion and *YPQ1* overexpression led to a slight reduction in lysine vacuolar fraction which is in line with a role of Vsb1 and Ypq1 in vacuolar compartmentalization.

We finally examined the epistasis relationship between *vsb1Δ* and *ypq1Δ* deletions by comparing the total lysine pool of the double *vsb1Δypq1Δ* mutant to those of the previously analyzed single mutant strains. The double mutant displayed similar soluble lysine levels as the *vsb1Δ* mutant (Fig 1A), indicating that the effect of *VSB1* deletion is epistatic over that caused by the *YPQ1* deletion. Thus, Vsb1 likely acts upstream of Ypq1 in the control of vacuolar lysine levels.

### Vsb1 attenuates Lyp1 down-regulation and mitigates lysine toxicity by mediating proton gradient dependent vacuolar accumulation of lysine

Next, we examined the role of Vsb1 in lysine homeostasis in the presence of exogenous lysine. The total free lysine content in the WT and the *vsb1Δ* strains was quantified before and after addition of 500 µM lysine. Although lysine supplementation increased intracellular lysine pools in both cases, the *vsb1Δ* cells showed lower total pool (8.5-fold less) as compared with the WT strain after 2 h (Fig 2A). One plausible explanation for this impairment is a defect in lysine import and accumulation in cells lacking Vsb1.

To assess lysine uptake directly, we measured the effect of the *VSB1* deletion on the kinetics of [14C]-lysine accumulation in the WT and the *vsb1Δ* strains over 60 min. Long-term accumulation of [14C]-lysine was strongly reduced in the *vsb1Δ* strain (Fig 2B). To determine whether this decrease stemmed from impaired activity of the plasma membrane lysine transporter Lyp1 (Jauniaux & Grenson, 1990; Sychrova et al, 1993), the initial uptake rates of [14C]-lysine were measured at 15 s interval over 1 min. The *vsb1Δ* mutant displayed a similar initial uptake rate to the WT (Fig 2C). These results indicate that Lyp1 activity decreases over time faster in the *vsb1Δ* mutant compared with the WT (Fig 2B). We, therefore, asked whether reduced lysine accumulation in *vsb1Δ* cells could be linked to Lyp1 down-regulation. To this end, we expressed, in WT and *vsb1Δ* strains, a C-terminally GFP-tagged Lyp1 from a plasmid which was validated as fully functional as it can transport [14C]-lysine (Fig S1A). In the absence of exogenous lysine, Lyp1-GFP localized predominantly at the plasma membrane in both strains (Fig 2D). However, upon the addition of external lysine, Lyp1-GFP was internalized, and this occurred significantly faster in the *vsb1Δ* mutant than in the WT cells (Fig 2D). In a hypomorphic *rsp5* mutant background, in which endocytosis of plasma membrane transporters is impaired as a consequence of reduced expression of the Rsp5 ubiquitin ligase (Hein et al, 1995), Lyp1-GFP internalization was blocked, confirming that it occurs via endocytosis. This suggests that faster internalization of Lyp1 in response to lysine addition leads to reduced [14C]-lysine accumulation in the *vsb1Δ* mutant. In accordance with this, introducing the *rsp5* mutation in a

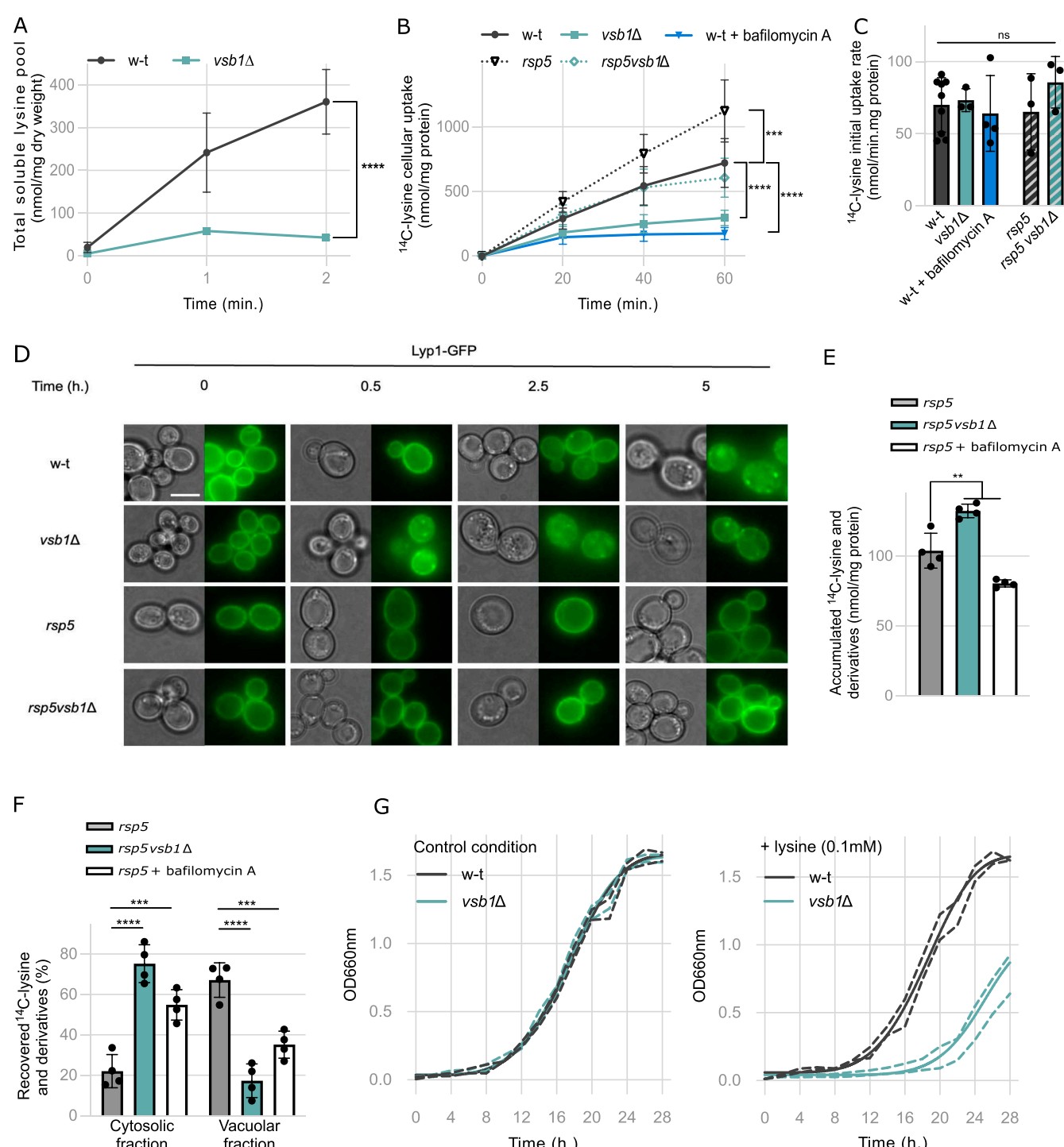

**Figure 2. Vsb1 attenuates Lyp1 down-regulation and mitigates lysine toxicity by mediating proton gradient dependent vacuolar accumulation of lysine.**
**(A)** The intracellular lysine content was measured in the WT and *vsb1Δ* strains 0, 1 and 2 h after addition of 500 $\mu$M lysine in the media. (****: $P < 0.0001$ by two-way ANOVA test) (n = 4 biological replicates). Error bars are included for all data points; however, they are not visible for the *vsb1Δ* mutant because they fall within the symbol size. **(B)** Time course accumulation of $^{14}$C-lysine (500 $\mu$M) over 60 min into WT, *vsb1Δ*, WT treated with bafilomycin A (9 $\mu$M) for 15 min before the assay, *rsp5* and *rsp5vsb1Δ* cells (***: $P < 0.0021$; ****: $P < 0.0001$ by two-way ANOVA) (n = 3–9 biological replicates). **(C)** Initial uptake rates of $^{14}$C-lysine (500 $\mu$M) in WT, *vsb1Δ*, WT treated with bafilomycin A (9 $\mu$M) for 15 min before the assay (90 $\mu$M $^{14}$C-lysine), *rsp5* and *rsp5vsb1Δ* cells (ns: $P > 0.05$ by one-way ANOVA) (n = 3–9 biological replicates). Initial uptake rates were determined within the first 60 s after $^{14}$C-lysine addition. **(D)** Microscopy analysis of Lyp1-GFP expressed from a plasmid in WT, *vsb1Δ*, *rsp5*, and *rsp5vsb1Δ* strains before and after 500 $\mu$M lysine addition in the culture media (0.5, 2.5, and 5 h). Scale: 5 $\mu$m. **(E)** The $^{14}$C-lysine accumulated in the *rsp5* and *rsp5vsb1Δ* strains (30 and 90 $\mu$M $^{14}$C-lysine, respectively) and the *rsp5* strain treated with bafilomycin A (9 $\mu$M) for 15 min before the assay (90 $\mu$M $^{14}$C-lysine) (**: $P > 0.0021$ by one-way ANOVA test) (n = 4 biological replicates). **(E, F)** Distribution of initially accumulated $^{14}$C-lysine and derivatives (from (E)) between the cytosolic and vacuolar fractions after cell permeabilization with cytochrome C (***: $P < 0.0002$; ****: $P < 0.0001$ by one-way ANOVA test) (n = 4 biological replicates). **(G)** Growth of the WT and *vsb1Δ* strains in the absence or presence of lysine (100 $\mu$M) in the culture media. Cells were initially grown on minimal lysine-free medium containing proline (10 mM) as the sole

WT strain or a *vsb1Δ* mutant led to a twofold higher accumulation of $^{14}$C-lysine in whole cells (Fig 2B).

These observations collectively imply that Vsb1 activity helps to maintain Lyp1 at the plasma membrane by mediating vacuolar lysine import, thereby preventing excessive cytosolic lysine accumulation. To test whether Vsb1 promotes vacuolar uptake of exogenous lysine, we incubated cells with $^{14}$C-labeled lysine, and then selectively permeabilized the plasma membrane using cytochrome C to successively extract cytosolic and vacuolar fractions (Wiemken & Nurse, 1973; Messenguy et al, 1980; Cools et al, 2020). We used the *rsp5* mutant background to enhance the accumulation of the radiolabeled compound and improve signal-to-noise ratio. In addition, the concentrations of $^{14}$C-lysine were adjusted to ensure comparable intracellular accumulation (Fig 2E). In the *rsp5* mutant, over 60% of $^{14}$C-lysine and its derivatives were vacuolar. In contrast, in *rsp5vsb1Δ* cells, the vacuolar content of $^{14}$C-lysine dropped to 20%, whereas ~80% was retained in the cytosol (Fig 2F). This confirms that Vsb1 contributes to the vacuolar import of exogenous lysine in growing cells and supports a model in which defective lysine import into the vacuole, as in the *vsb1Δ* mutant, leads to an excessive accumulation of lysine in the cytosol that triggers an early endocytosis of Lyp1.

To test whether Vsb1 activity is dependent on the vacuolar proton gradient, we measured the effect of bafilomycin A on the kinetics of $^{14}$C-lysine accumulation in WT cells. The initial lysine uptake rate of the treated cells remains similar to the uptake rate of the untreated cells (Fig 2C), indicating that Lyp1 activity (Jauniaux & Grenson, 1990; Sychrova et al, 1993) remained unaffected. However, long-term accumulation of $^{14}$C-lysine was strongly reduced upon bafilomycin A treatment (Fig 2B). In addition, we verified if lysine transport into the vacuole is dependent on the V-ATPase activity by treating *rsp5* cells with bafilomycin A before the addition of $^{14}$C-lysine and permeabilization. We again adjusted exogenous lysine concentrations to reach comparable intracellular accumulation (Fig 2E). Treating cells with bafilomycin A resulted in a pronounced shift of $^{14}$C-lysine and its derivatives towards the cytosolic fraction, mirroring the *rsp5vsb1Δ* (Fig 2F). Both results are in line with the hypothesis that Vsb1-dependent lysine accumulation into the vacuole requires the proton gradient established by the V-ATPase.

Finally, to assess the physiological relevance of Vsb1-mediated lysine compartmentalization in growing cells, we investigated how Vsb1 could contribute to protecting cells from lysine toxicity on poor nitrogen sources (Sumrada, 1976). To this end, we monitored the growth of WT and *vsb1Δ* strains in proline medium with or without lysine supplementation. Supplementation with 100 $\mu$M lysine did not affect the growth of the WT strain (Fig 2G). In contrast, the *vsb1Δ* mutant exhibited a longer lag phase and a reduced growth rate in the presence of excessive lysine. When ammonium was used as a nitrogen source, the hypersensitivity of the *vsb1Δ* strain to lysine was less pronounced (Fig S1B), likely because the Gap1 permease involved in lysine uptake is less active under these conditions (Grenson, 1983; Merhi & Andr, 2012). These observations

support the importance of vacuolar accumulation of lysine mediated by Vsb1 in maintaining lysine homeostasis and protecting cells from its toxic effects.

### The cytosolic RmlC-like domain of Vsb1 is essential for its CAA transport activity

To gain further insight into the mechanism of Vsb1 function, we set out to investigate the structural basis of its CAA transport function. Evolutionarily, Vsb1 belongs to the SLC26A/SulP superfamily of anion exchangers or anion channels ubiquitous in all kingdoms of life (Alper & Sharma, 2013). According to several available structures, SLC26A/SulP transporters contain a transmembrane domain having 14 α-helices and a STAS domain required for dimerization. In the case of Vsb1, it has a cytosolic N-terminal tail and a putative RmlC-like domain in addition to the SLC26A/SulP-like transmembrane domain and the STAS domain (Fig 3A). Because the structure of Vsb1 has not yet been solved, we modeled its molecular architecture as a dimer using the AlphaFold 3 Server (Abramson et al, 2024). The full-length homodimer has a predicted template modeling score of 0.59, indicating overall poor structure prediction (Fig S2A). However, several regions of the protein appeared disordered with a predicted confidence score below 0.5 (residues 1–188 and 818–848), indicating that either these regions are natively unfolded or fail to be properly modeled because of the lack of structural homologs in the Protein Data Base (Terwilliger et al, 2024). Indeed, when these poorly modeled regions are omitted, the Vsb1 model exhibited a much better confidence score, between 0.7 and 0.9, compatible with its functional interpretation (Fig S2A).

As expected, the predicted folds of the transmembrane domain (195–638) and STAS domain (639–772) are closely related to the established SLC26A/SulP transporter structures. The former is composed of 14 TM α-helices segregated into a gate and a core subdomain (Fig S2B). The putative substrate binding pocket is lodged between TM3 and TM10 (Figs 3B and S2B). The predicted interaction between the monomers in the dimer model is mediated partially by the STAS domains with an interaction surface of 385 $\text{Å}^2$. Since the STAS domains are swapped between monomers, the STAS domain of a monomer seems to also interact with the transmembrane domain of the other monomer (Fig 3A). Finally, the fourth domain of Vsb1 (773–1,036) is predicted to adopt an RmlC-like jelly roll fold (849–1,036) and to connect to the STAS domain by a long junctional α-helix (774–848). Such a fold is often associated with proteins that bind cyclic nucleotide, nucleotide-activated sugar, or metal ion (Dunwell et al, 2001). The RmlC-like domain can also contribute to protein multimerization as observed for the mammalian potassium channel Eag1 (Whicher & MacKinnon, 2016). In Vsb1, it could possibly play a role in dimerization because it provides a massive projected interaction surface between monomers of 2,180 $\text{Å}^2$. The presence of such a domain in a transporter has only been reported for LtnT, a SLC26A/SulP transporter involved in nitrate transport from the cyanobacterium

nitrogen source and the optical density at 660 nm (OD 660 nm) was monitored over a 24-h period. Growth is represented as the Weibull nonlinear fit curve with the 99% confidence bands (n = 3 biological replicates).

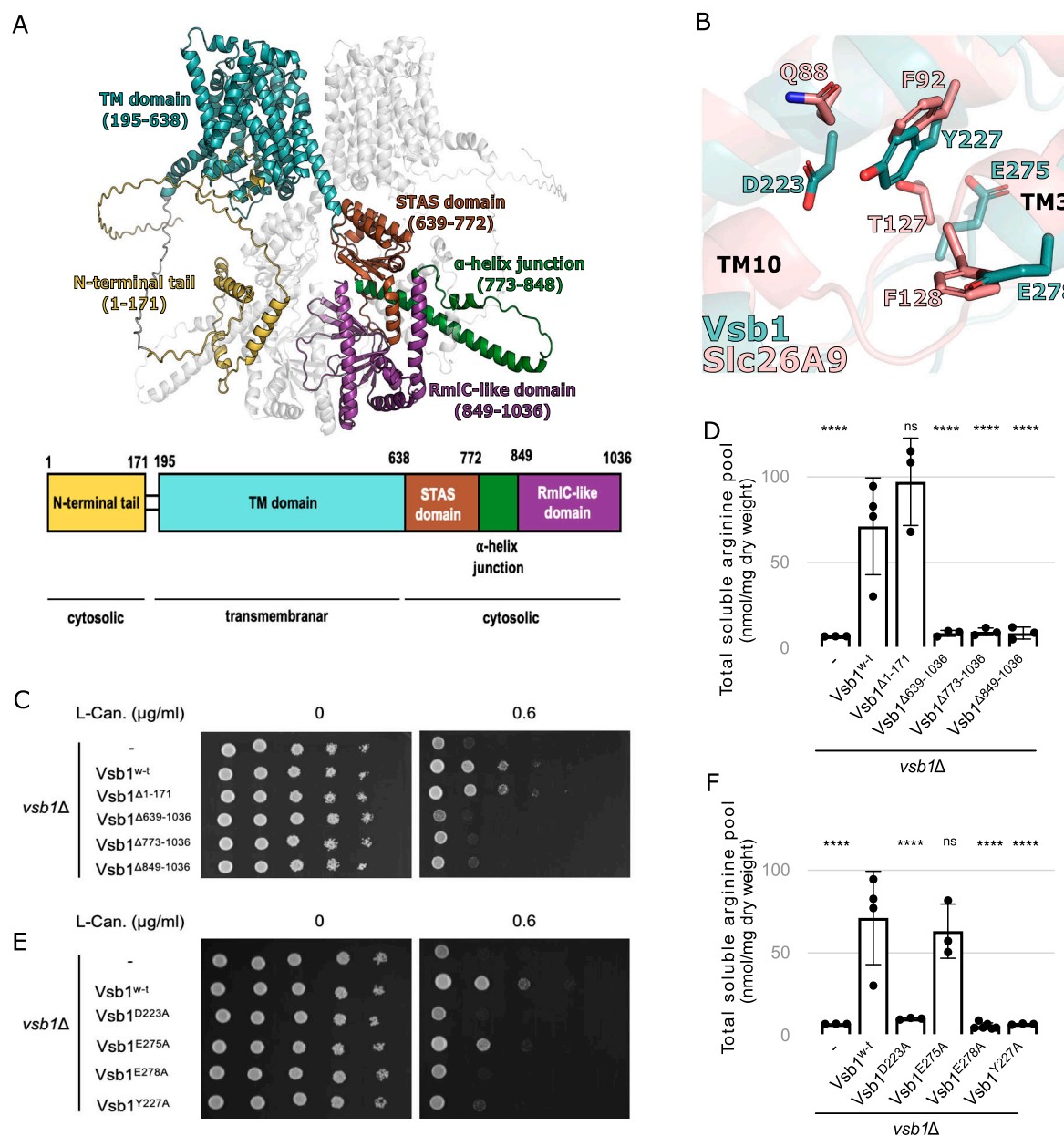

**Figure 3. Insights into Vsb1 function from structural modeling and mutational analyses.**
**(A)** Overall representation of Vsb1 structure modeled as a dimer using AlphaFold 3. The domains and regions of monomer A are represented as: N-terminal tail in yellow, the transmembrane (TM) domain in teal, the STAS domain in red, the α-helix junction in green, and the RmlC-like domain in purple. Monomer B is represented in light grey. A schematic representation of Vsb1 domain architecture is given with the predicted cellular sublocalization. **(B)** Close-up view of the putative CAA binding site of Vsb1 (teal) and comparison with the Cl⁻ binding site of the mammalian Slc26A9 (pink). The residues involved in chloride transport in Slc26A9 (Walter et al, 2019) and their Vsb1 counterparts are annotated and shown in stick representation. The R.M.SD. of the structural alignment Vsb1 model with Slc26A9 (PDB code 6RTC) is 3.76 Å. **(C)** The effect of Vsb1 truncations was evaluated by assessing canavanine susceptibility of a *vsb1Δ* strain transformed with an empty plasmid (−) or a plasmid allowing the expression of different *VSB1* truncation mutants under the *VSB1* native promoter. Cells were spread on minimal medium containing 0.6 µg/ml canavanine or not and cell growth was evaluated after a 3-d incubation at 29°C. **(D)** The intracellular arginine content was measured for the *vsb1Δ* strains expressing the different *VSB1* truncations (ns: $P > 0.05$; ****: $P < 0.0001$ by one-way ANOVA with post hoc comparison tests with the WT strain) (n = 3–4 biological replicates). **(E, F)** A *vsb1Δ* strain was transformed with an empty plasmid (−) or plasmids expressing different *VSB1* mutants under the *VSB1* native promoter. **(E, F)** Cells spread on minimal medium containing canavanine or not were incubated for 3 d at 29°C (E) and their intracellular arginine content was measured (F) (ns: $P > 0.05$; ****: $P < 0.0001$ by one-way ANOVA with post hoc comparison tests with the WT strain) (n = 3–4 biological replicates).

*Synechococcus elongatus* (Maeda et al, 2006). It is worth noticing that Vsb1 lacks the intervening sequence and the PDZ domain found in mammalian SLC26A/SulP transporters.

To investigate the functional significance of the cytosolic domains, we generated Vsb1-truncation variants that encompass the N-terminal tail (Vsb1$^{Δ1–171}$), the RmlC-like domain (Vsb1$^{Δ849–1,036}$),

the RmlC-like domain with the α-helix junction (Vsb1$^{\Delta773-1,036}$), or the entirety of the C-terminal region (Vsb1$^{\Delta639-1,036}$) (Fig 3A). First, we analyzed their intracellular localization using the C-terminal GFP fusions expressed under the *VSB1* native promoter in a *vsb1Δ* deletion background. The truncation variants were expressed mainly as full-length proteins although at somewhat different levels (Fig S3B). Most of them were also efficiently targeted to the vacuolar membrane as shown by colocalization with lipophilic dye FM4-64 used as a fiducial vacuolar membrane marker (Vida & Emr, 1995) (Fig S3A). To test the functionality of the truncated variants, we expressed them in an untagged form in the *vsb1Δ* deletion background, as GFP fusion was shown to be incompatible with Vsb1 activity (Cools et al, 2020; Kawano-Kawada et al, 2021), and tested them for supporting growth on plates containing canavanine, an analogue of arginine to which yeast cells lacking *VSB1* are hypersensitive (Cools et al, 2020) (Fig 3C). Whereas complementing *vsb1Δ* mutant with Vsb1$^{\Delta1-171}$ supported cell growth in the presence of canavanine similarly to WT Vsb1, the other truncation variants failed to complement the *vsb1Δ* mutant. Consistent with these observations, the same truncations impaired the intracellular accumulation of CAA (Figs 3D and S3D and E). Together these results indicate that the cytosolic C-terminal domains are essential for Vsb1 activity and a critical part can be mapped to the RmlC-like domain (residues 849–1,036) (Fig 3C and D).

### Asp-223, Tyr-227, and Glu-278 of the putative CAA binding site are essential for Vsb1 activity

Next, we sought to investigate the requirement of specific residues within the transmembrane domain of Vsb1 for CAA transport, explicitly because members of the SLC26A/SulP superfamily are classically described as anion transporters. In mammalian SLC26A transporters, the anion binding site is globally positively charged via the macro-dipoles of TM3 and TM10 and within this site an arginine residue plays a crucial role in anion binding for SLC26A3, SLC26A4, SLC26A5, and SLC26A6 . Therefore, there must be some adaptations in Vsb1 to allow the binding of CAA. A previous study has identified Asp-223, which is conserved among the fungal Vsb1 orthologues, as being necessary for transport of CAA (Kawano-Kawada et al, 2021; Ohnishi et al, 2022). Based on our model, Asp-223 is conveniently located in the putative binding site between TM3 and TM10 (Fig 3B). In the characterized mammalian transporters of SLC26A/SulP family, a glutamine residue is found at a position equivalent to Asp-223 and implicated in anion interaction (Figs 3B and S2C). In SLC26Dg from the bacterium *Deinococcus geothermalis,* a glutamate residue equivalent to Asp-223 is required for the fumarate/Na$^+$ exchange (Geertsma et al, 2015). By comparing the Vsb1 model with the related SLC26A transporters, we identified three other candidate residues that could be involved in CAA transport: Tyr-227, Glu-275, and Glu-278 (Figs 3B, S2C, and S4). Of note, the arginine residue found in the anion binding site of some SLC26A/SulP transporters is absent in Vsb1 (Fig S2C and D). In all structurally characterized SLC26A members, an aromatic residue is found within the substrate binding site at a position equivalent to Tyr-227 (Fig S2C) and plays a role in anion exchange (Butan et al, 2022; Futamata et al, 2022; Tippett et al, 2023; Hu et al, 2024). Both Asp-223 and Tyr-227 are within a conserved

region of the SLC26A members (Fig S4). Glu-275 is positioned in the second shell of interaction, close to Tyr-227, possibly imposing steric constraint on the tyrosyl group. Finally, Glu-278 is localized at the N-terminal end of TM3 and could neutralize the helix macro-dipole. It has been proposed that the helix macro-dipoles of TM3 and TM10 play a role in anion binding in some SLC26A transporters (Geertsma et al, 2015; Tippett et al, 2023).

To investigate the roles of these four candidate key residues, Vsb1 variants were constructed, each of them separately substituted with an alanine residue. As in the case of truncations, the GFP-fused mutants correctly localized to the vacuolar membrane and were expressed at similar levels to the WT variant (Fig S3A and C). The untagged variants were then tested for their ability to rescue the canavanine hypersensitivity (Fig 3E) and reduced levels of CAA of the *vsb1Δ* deletion mutant (Figs 3F and S3E). Whereas Vsb1$^{E275A}$ variant was only slightly more sensitive to canavanine than the WT and had levels of cellular CAA similar to the WT cells, the Vsb1$^{D223A}$, Vsb1$^{Y227A}$, and Vsb1$^{E278A}$ variants failed to rescue both phenotypes. These observations strongly support the critical role of Asp-223, Tyr-227, and Glu-278 in CAA transport by Vsb1.

### Ypq1 is essential for lysine mobilization from the vacuole under lysine starvation

Our observation that lysine accumulates to higher levels in the *ypq1Δ* mutant and that its overexpression reduced the lysine pool (Fig 1A) is consistent with Ypq1 acting as a vacuolar lysine exporter. To clarify the role of Ypq1 in vivo, we first assessed the activation of the lysine biosynthesis pathway in both WT and *ypq1Δ* strains after lysine withdrawal, reasoning that this response could serve as an indicator of the availability of cytosolic lysine. To monitor this response, cells were grown in a lysine-supplemented medium and then washed and resuspended in a lysine-free medium when monitoring mRNA levels of *LYS9*, the gene that encodes saccharopine dehydrogenase involved in lysine biosynthesis and that is known to be strongly induced upon lysine deprivation (Storts & Bhattacharjee, 1987). In the presence of lysine, the expression of *LYS9* was comparable in both WT and *ypq1Δ* strains and, as expected, was induced after lysine withdrawal (Fig 4A). However, in the *ypq1Δ* mutant, this induction occurred earlier and was much more pronounced. This increased biosynthetic response is consistent with the view that lysine is released from the vacuole less efficiently in the absence of Ypq1.

To test the role of Ypq1 in the vacuolar export of lysine more directly, we compared total soluble lysine pools of *lys2Δ* and *lys2Δypq1Δ* strains subjected to lysine starvation. Here the *LYS2* gene was deleted to confer lysine auxotrophy (Sinha & Bhattacharjee, 1971; Feller et al, 1999). The WT, *lys2Δ* and *lys2Δypq1Δ* strains were initially cultured in minimal medium supplemented with 500 µM lysine and then shifted to lysine-free medium. In the presence of external lysine, all three strains accumulated lysine at similar levels (Fig 4B). However, upon lysine withdrawal, total lysine pools rapidly declined in the WT and the *lys2Δ* mutant strains but not in *lys2Δypq1Δ* double mutant which still retained ~65% of its initial lysine content even after 6 h of starvation (Fig 4C). We. therefore, conclude that lysine mobilization

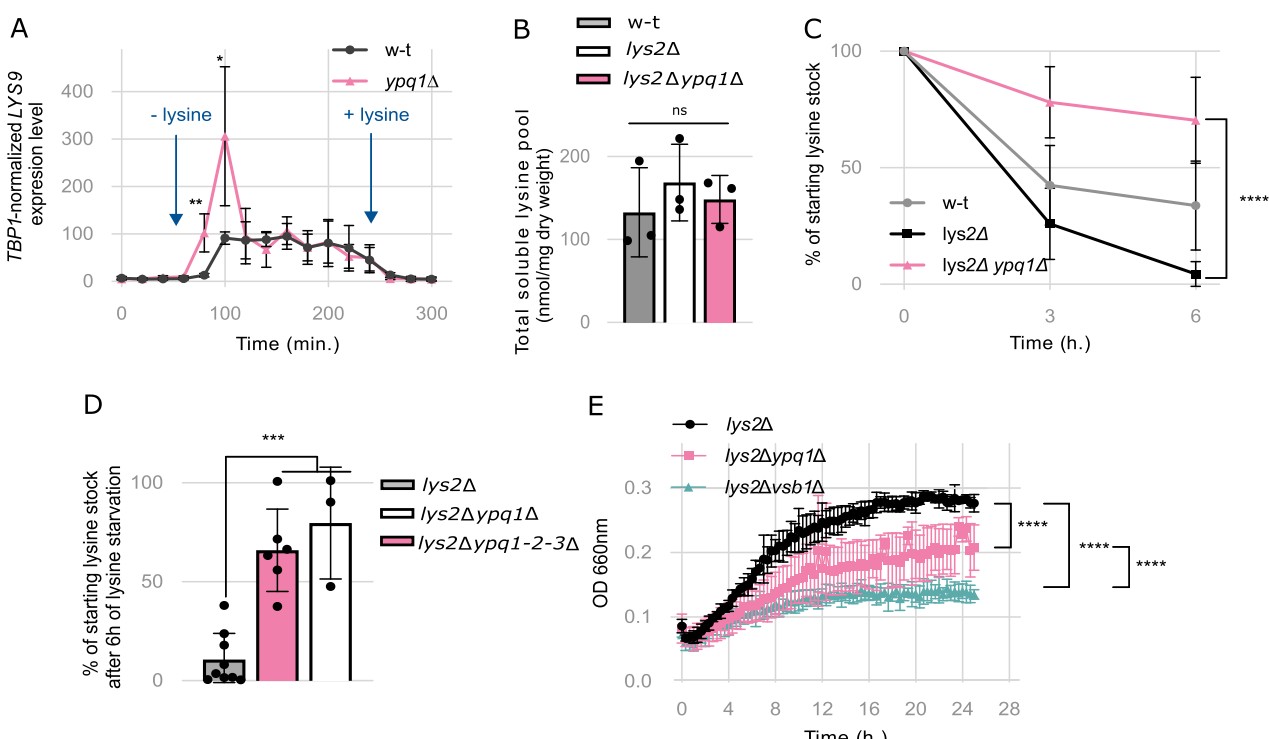

**Figure 4. Ypq1 is essential for lysine mobilization from the vacuole under lysine starvation.**
**(A)** Impact of lysine on relative expression levels of *LYS9* in the WT and *ypq1Δ* strains. Cells were grown in minimal medium supplemented with lysine (500 µM). At the designated time points, cells were collected, washed and resuspended in a lysine-free minimal medium (–lysine), followed by re-supplementation with lysine (+lysine). Samples were collected every 20 min over a 5 h period for quantitative RT–PCR analysis of *LYS9* expression (*: $P < 0.05$; **: $P < 0.005$ by *t* test) (n = 4 biological replicates). **(B)** The intracellular lysine content was measured in the WT, *lys2Δ* and *lys2Δypq1Δ* strains grown in minimal medium supplemented with lysine (500 µM) (ns: $P > 0.05$ by one-way ANOVA test) (n = 3 biological replicates). **(C)** Percentage of the initial intracellular lysine pool left after lysine starvation. The intracellular lysine content in the WT, *lys2Δ*, and *lys2Δypq1Δ* strains was measured 0, 3, and 6 h after transfer from a minimal medium supplemented with lysine (500 µM) to a lysine-free medium (****: $P < 0.0001$ by one-way ANOVA test) (n = 4–12 biological replicates). **(D)** The intracellular lysine content in the *lys2Δ*, *lys2Δypq1Δ*, and *lys2Δypq1-2-3Δ* strains was measured 0 and 6 h after transfer from a minimal medium supplemented with lysine (500 µM) to a lysine-free medium. (***: $P < 0.001$ by one-way ANOVA test) (n = 3–9 biological replicates). **(E)** Residual growth of the *lys2Δ*, *lys2Δypq1Δ*, and *lys2Δvsb1Δ* strains under lysine starvation. Cells were initially grown on minimal medium supplemented with lysine (416 µM) then collected by centrifugation and washed with lysine-free medium. Subsequently, cells were resuspended in lysine-free medium and the optical density (OD 660 nm) was monitored over a 24-h period (****: $P < 0.0001$ by two-way ANOVA test) (n = 3 biological replicates).

from the vacuole is impaired in the *lys2Δypq1Δ* mutant, indicating that Ypq1 mediates vacuolar lysine export under lysine scarcity.

Because the *lys2Δypq1Δ* mutant was still able to mobilize 35% of its soluble lysine stores, other transporters could contribute to the export of lysine from the vacuole. Obvious candidates for this role include Ypq2 and Ypq3, two proteins evolutionarily related to Ypq1, which have been previously described as CAA transporters (Jézégou et al, 2012; Manabe et al, 2016; Kawano-Kawada et al, 2019; Cools et al, 2020). To determine whether Ypq2 or Ypq3 can mobilize lysine from the vacuole, we, in addition, analyzed the lysine stores of the *lys2Δypq1-2-3Δ* strain after 6 h of lysine starvation. However, the quadruple mutant did not show any significant difference compared with the *lys2Δypq1Δ* strain (Fig 4D), indicating that neither Ypq2 nor Ypq3 effectively contribute to lysine mobilization in these conditions.

We next examined whether impaired export of lysine from the vacuole or defective accumulation before starvation affects the ability of the cell to sustain proliferation. We thus compared the proliferation capacity of *lys2Δ*, *lys2Δypq1Δ*, and *lys2Δvsb1Δ* strains following lysine withdrawal. The residual growth of the

*lys2Δypq1Δ* mutant, which is defective in lysine mobilization, and the *lys2Δvsb1Δ* mutant, which stores lysine at much lower levels, was markedly reduced under lysine starvation compared with the *lys2Δ* strain (Fig 4E). This suggests that vacuolar lysine mobilization mediated by Ypq1 is an important means to maintain cell growth under lysine starvation.

Previous studies have reported vacuolar internalization and degradation of Ypq1 in response to lysine starvation (Li et al, 2015b; Arines et al, 2021). Considering our findings implicating Ypq1 in vacuolar lysine export during lysine starvation, we set out to investigate whether Ypq1 is degraded in our experimental conditions. To this end, Ypq1 was endogenously C-terminally tagged with GFP (Ypq1-GFP) in the WT and the *lys2Δ* mutant. The Ypq1-GFP construct was functional as evidenced by the WT levels of soluble intracellular lysine (Fig S5A) and by the expected targeting of the GFP fusion to the vacuolar membrane observed in both strains grown in the presence of lysine (Fig S5B and C). When cells were transferred to a lysine-free medium, Ypq1-GFP remained at the vacuolar membrane in the WT strain, even after 6 h. In contrast, three distinct cell populations were observed in the *lys2Δ* mutant:

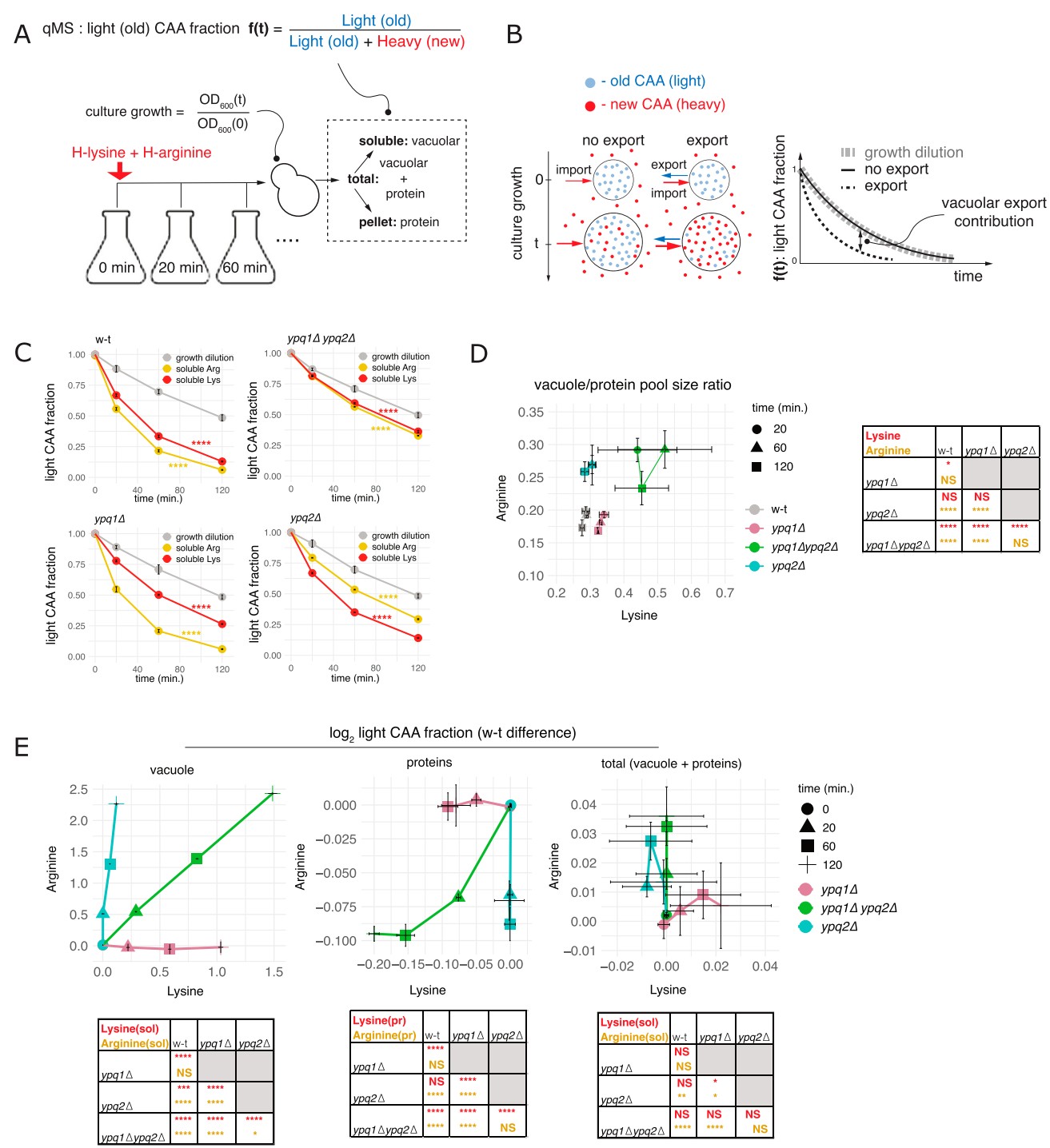

**Figure 5. Ypq1 and Ypq2 mediate transport of lysine and arginine in actively growing cells.**
**(A)** Schematic of a dynamic labeling assay to monitor CAA renewal dynamics and cell culture growth. Yeast cultures grown in SCD medium containing normal (light) lysine and arginine were transferred by filtration in the mid-log phase (OD 600 nm ~0.15) to a pre-warmed "heavy" SCD medium containing the same concentrations of $[^{13}C_6, ^{15}N_2]$-lysine and $[^{13}C_6, ^{15}N_4]$-arginine. At each time point after the labeling onset, the culture samples were drawn, part to determine cell culture growth using turbidity (OD 600 nm) and part for CAA quantification in subcellular fractions. For CAA quantification cells were washed with distilled water, and the boiled cell suspension (total CAA pool) was split by centrifugation into soluble fraction and pellet representing vacuolar and protein-borne CAA, respectively. The insoluble and total fractions were additionally hydrolyzed for 24 h with 6 M HCl at 110°C to release protein-borne CAA. All three fractions were analyzed by quantitative mass spectrometry to determine the fractional content of the light CAA variants, referred to as renewal or labeling. **(B)** Renewal dynamics of vacuolar CAA and its connection to vacuolar export. In the absence of export, the vacuolar pool of "old" (light) CAA present at labeling onset (time = 0, blue) can only be diluted by the import of "new" (heavy) CAA (time = t, red) to support cell culture growth ("no export" scenario). However, observing faster renewal dynamics implies higher CAA import than necessary for sustaining cell culture growth that requires some vacuolar export to maintain balanced growth. Graph on the right illustrates renewal dynamics of vacuolar CAA (light CAA fraction) corresponding to both scenarios. **(C, D, E)** Readouts of dynamic labeling assays in WT, ypq1Δ, ypq2Δ, and ypq1Δypq2Δ strains. **(C)** Renewal dynamics of the vacuolar CAA as compared with growth dilution dynamics evaluated for the same cultures as inverse of cell culture growth. (****: difference between log-transformed

one in which Ypq1-GFP persisted at the vacuolar membrane, another in which it was targeted to the vacuolar lumen and a third in which an intermediate condition was observed (Fig S5B and C). To verify that luminal GFP was associated with the degradation of Ypq1-GFP, cell extracts were collected under the same conditions and analyzed by western blotting (Fig S5D and E). Following lysine withdrawal, Ypq1-GFP levels in the *lys2Δ* mutant showed a slight decrease, accompanied by an accumulation of free GFP (Fig S5D and E). Notably, this did not occur in the lysine-prototrophic strain. These results confirm previous reports that Ypq1 is down-regulated upon lysine starvation (Li et al, 2015b; Zhu et al, 2017; Arines et al, 2021), although this response appears to be partial and heterogeneous across the cell population. Interestingly, Ypq1 down-regulation occurs several hours after lysine withdrawal, when less than 20% of the initial soluble lysine stock remains (Fig 4C), suggesting that Ypq1 down-regulation is connected to the depletion of its intravacuolar substrate.

All in all, these results indicate that Ypq1 is the major vacuolar lysine exporter, necessary to mobilize vacuolar lysine stores under lysine starvation conditions, dynamically regulated in correlation with vacuolar lysine availability.

### Lysine and arginine are exported from the vacuole during active cell growth

Because *YPQ1* and *YPQ2* deletions lead to cellular over-accumulation of lysine (Fig 1A) and arginine (Cools et al, 2020), respectively, CAA could be trapped in the vacuoles of these mutants because of decreased export. We therefore were interested to understand whether CAA are normally exported from the vacuole during active cell growth and, if so, to what extent export of lysine and arginine depends on Ypq1 and Ypq2. We thought to address these questions directly in live cells by a dynamic labeling approach in which steady-state growing yeast cultures are switched from a medium containing normal (light) lysine and arginine to an equivalent medium with their heavy stable isotope analogs, so that one can follow the content of *old* and *new* CAA encoded by the light and heavy isotopologues using quantitative mass-spectrometry (Fig 5A). We reasoned that this approach would allow us to detect vacuolar CAA export (or lack thereof) if we simultaneously monitor and compare dynamics of cell culture growth with the dynamics of vacuolar CAA renewal, measured as fractional content of light amino acids in the vacuole, after the medium switch (Fig 5A). Indeed, if the steady-state culture grew for one generation since the medium switch, the vacuolar mass is expected to double. Should CAA only get into the vacuole but do not get out, this will result in the best case in 1:1 dilution of the old/light amino acid in the vacuolar pool with the new/heavy counterpart supplied from the exchanged medium. In other words, in the absence of export the CAA pool cannot be renewed faster than dilution due to culture growth (Fig 5B, left). However, observing CAA

renewal exceeding the growth dilution would necessarily imply that additional import into the vacuole must have occurred, beyond what is required to keep up with the culture growth, which is only possible if the CAA were exported from the vacuole to compensate for this additional import (Fig 5B, right). Otherwise, this additional influx would lead to faster growth of the vacuolar mass compared with other cellular parts, which is not compatible with the steady-state growth conditions.

To implement this analysis framework, we require that the light (i.e., old) CAA molecules not be contaminated with the CAA produced de novo after the medium switch. We must also ensure that, when simultaneously monitoring cell culture growth and CAA renewal dynamics, cell cultures maintain steady-state exponential growth and the cellular vacuolar CAA pools do not change compared with other cellular parts, that is, the growth remains balanced. To ensure that both lysine and arginine are not produced internally, we used *lys2Δarg4Δ* background strain, in which biosynthesis of both amino acids cannot occur (Sinha & Bhattacharjee, 1971; Beacham et al, 1984; Ljungdahl & Daignan-Fornier, 2012). To simultaneously monitor growth dynamics and amino acid renewal, cell culture samples were drafted at certain time points after the medium switch and used part to measure culture turbidity and part to quantify CAA by quantitative mass spectrometry. Turbidity measurements allowed us to compute biomass growth dynamics (i.e., how much cell culture grew since the labeling onset) and were used to ensure exponential growth and to determine growth dilution dynamics (see the Materials and Methods section). Using quantitative mass spectrometry, in addition to vacuolar pool we aimed to accurately determine renewal dynamics of CAA (i.e., their light fraction) also in the protein pool and in both these pools combined (Fig 5A). This was necessary to ensure balanced vacuolar growth and, later, for the detailed analysis of vacuolar CAA transport. To this end, we have analyzed CAA renewal in total cell lysate and in its soluble and insoluble fractions, considering that they represent the aggregate CAA pools (vacuolar + protein) and the vacuolar and protein pools separately (Fig 5A). We have ensured that in *lys2Δarg4Δ* mutant cells soluble CAA are mainly vacuolar, i.e., that the soluble fraction can be used as proxy for the vacuolar CAA pool, by analyzing the soluble CAA partition between the vacuolar and cytosolic fractions with $Cu^{2+}$ permeabilization and phenylalanine as a fiducial cytosolic marker (Messenguy et al, 1980; Kitamoto et al, 1988) (see the Materials and Methods section). This showed that more than 95% of both lysine and arginine in *lys2Δarg4Δ* cells appeared vacuolar (Fig S7A). Because all cellular CAA are essentially divided between vacuolar (soluble) and protein (insoluble) pools, we could use their renewal quantified in all three fractions to accurately determine relative content in the vacuole, vacuole-to-protein pool size ratio (see the Materials and Methods section) and use it as a criterion of balanced vacuolar growth.

renewal dynamics and growth dilution $P < 0.0001$ by two-way ANOVA). **(D)** Vacuole-to-protein pool size ratios determined for CAA at different time points after labeling onset based on dynamic labeling readouts (see Materials and Methods) (****: $P < 0.0001$, *: $P < 0.05$ by two-way Tukey HSD). **(E)** Log-transformed renewal of CAA in the vacuolar, protein and total cellular pools measured in *ypq1Δ*, *ypq2Δ*, and *ypq1Δypq2Δ* strains and offset by the corresponding mean values in the WT strain (****: $P < 0.0001$, ***: $P < 0.001$, *: $P < 0.05$ by two-way Tukey HSD). Error bars (SD) and *P*-values are evaluated based on n = 3 biological replicates.

Having established the dynamic labeling framework, we asked whether cells export CAA from the vacuole during active growth in CAA replete conditions by measuring growth dynamics and CAA renewal in *lys2Δarg4Δ* cells over 2 h after the medium switch (Figs S6B and S7B). In these experiments cell cultures maintained exponential growth over the whole 2-h labeling time-course as judged by the linearity of log-transformed biomass growth dynamics ($R^2 > 0.99$) (Fig S7B). Strikingly, comparing the vacuolar CAA renewal with the growth dilution showed that both lysine and arginine were renewed faster than expected from growth dilution alone ($P < 0.0001$ by two-way ANOVA) (Fig 5C). To exclude that faster renewal was because of unbalanced growth, we computed vacuole-to-protein pool size ratios along the labeling time-course. If enhanced renewal was because of a sudden increase in CAA import, we expected to observe a disproportional growth of the vacuolar CAA pools compared to proteins. However, the vacuole-to-protein ratios remained highly similar for both lysine and arginine, thus excluding this possibility (Fig 5D). Thus, we conclude that actively growing yeast cells export both lysine and arginine from the vacuole under CAA replete conditions.

### The deletions of Ypq1 and Ypq2 semi-selectively affect the intracellular traffic of lysine and arginine

We next thought to use our dynamic labeling framework to investigate the roles of Ypq1 and Ypq2. We reasoned that if Ypq1 and Ypq2 mediate the vacuolar export of a CAA, their deletions will inevitably affect the dynamics of CAA renewal in the vacuolar and protein pools as two connected CAA stores, which can be detected by the labeling dynamics readouts. Following these premises, we delete the *YPQ1* and *YPQ2* genes alone or both genes together in the *lys2Δarg4Δ* background and subjected these strains to the same dynamic labeling analysis as WT cell. Like in the WT strain, we observed that the soluble CAA pools in the mutants were almost entirely vacuolar (Fig S7A). During dynamic labeling, mutant cell cultures also maintained exponential growth (Fig S7B) and did not show systematic changes in the size of the vacuolar CAA pool along the labeling time-course, all indicative of steady-state growth (Fig 5D). To highlight effects on renewal dynamics, we initially plotted renewal values (as log-scaled light CAA fraction) for lysine and arginine against each other for all four genetic backgrounds and all labeling time points (Fig S7C). Whereas the mutations did not show a noticeable effect in total cellular pools, the renewal effects were obvious in vacuolar and protein pools taken separately (Fig S7C). Specifically, while mutations appeared to reduce the renewal dynamics of lysine and arginine in the vacuolar pool, both vacuolar fractions renewed significantly faster than the growth dilution in all mutants ($P < 0.0001$ by two-way ANOVA) (Fig 5C). We, therefore, conclude that Ypq1 and Ypq2 affect vacuolar transport of CAA. However, even in the absence of both Ypq1 and Ypq2, actively growing yeast cells maintain some vacuolar CAA export capacity.

To explore the effect of the mutations in more detail, we offset all renewal values obtained in the mutants by the corresponding values in the WT cells (Fig 5E). Positive readings in these representations indicate slower renewal as compared to the WT cells. Similarly, whether the corresponding time points in one of the mutants are shifted higher or lower compared with another mutant indicates whether the renewal in the first is slower or faster than in the second. Statistical significance of these differences was tested within each fraction using two-way Tukey HSD applied to lysine and arginine each in all genetic conditions. Most strikingly, slower renewal of lysine and arginine in the vacuolar pools was always accompanied by faster renewal in the protein pool (Fig 5E). The balance of these effects was supported by smaller mutation-related effects in the total lysates that represent both pools combined as compared with the individual pools (Fig 5E). This indicates that the mutations are unlikely to affect the transport of CAA from the outside environment or their intracellular metabolism but alter their traffic between the vacuolar and protein pools. Furthermore, the effects of *YPQ1* and *YPQ2* deletions on lysine and arginine renewal were rather selective. The single deletion of *YPQ1* strongly impaired the renewal of vacuolar lysine, but not arginine, whereas the opposite was true for the single deletion of *YPQ2*. This supports the view that Ypq1 and Ypq2 are rather selective mediators of lysine and arginine transport, respectively. At the same time, the effects of the mutations were not completely independent. The renewal of vacuolar lysine was, in addition, reduced in *ypq1Δypq2Δ* cells compared with the *ypq1Δ* mutant (Fig 5E), and a similar synthetic effect was observed in the case of the size of the vacuolar lysine pool increased in *ypq1Δypq2Δ* compared with *ypq1Δ* cells (Fig 5D). These synthetic effects can potentially be explained by incomplete selectivity of Ypq2 towards arginine, being able also to partially support lysine transport.

### Ypq1 and Ypq2 are efficient mediators of the vacuolar export of lysine and arginine

The fact that *YPQ* deletions impair the renewal dynamics of the vacuolar CAA pools and have the opposite effect on the protein pool (Fig 5E) can potentially be explained if Ypq1 and Ypq2 were mediating vacuolar export of CAA. Indeed, we expect that stopping export by their gene deletions would not only impair the renewal dynamics of the respective CAA in the vacuole but also render old vacuolar CAA stores inaccessible for protein biosynthesis, effectively enhancing renewal of the protein pool (Fig S7D). We, therefore, thought to take advantage of our detailed measurements of the renewal dynamics to quantitatively assess the vacuolar export rates and compare them between the mutants. We reasoned that, because in steady-state growth conditions of our experiments the vacuolar import must exactly balance the export from the vacuole and the growth of the vacuolar mass because of cell culture growth, we could always determine fractional vacuolar export rates (i.e., vacuolar efflux in units of vacuolar pool size) if we knew the corresponding fractional import rates (the influx into the vacuole in units of vacuolar pool size) and cell culture growth rate, as the difference between the two (Fig 6A). Furthermore, vacuolar export rates can be compared between mutants by adjusting them for differences in the vacuolar pool size using the determined vacuolar-to-protein pool size ratios (Fig 5D).

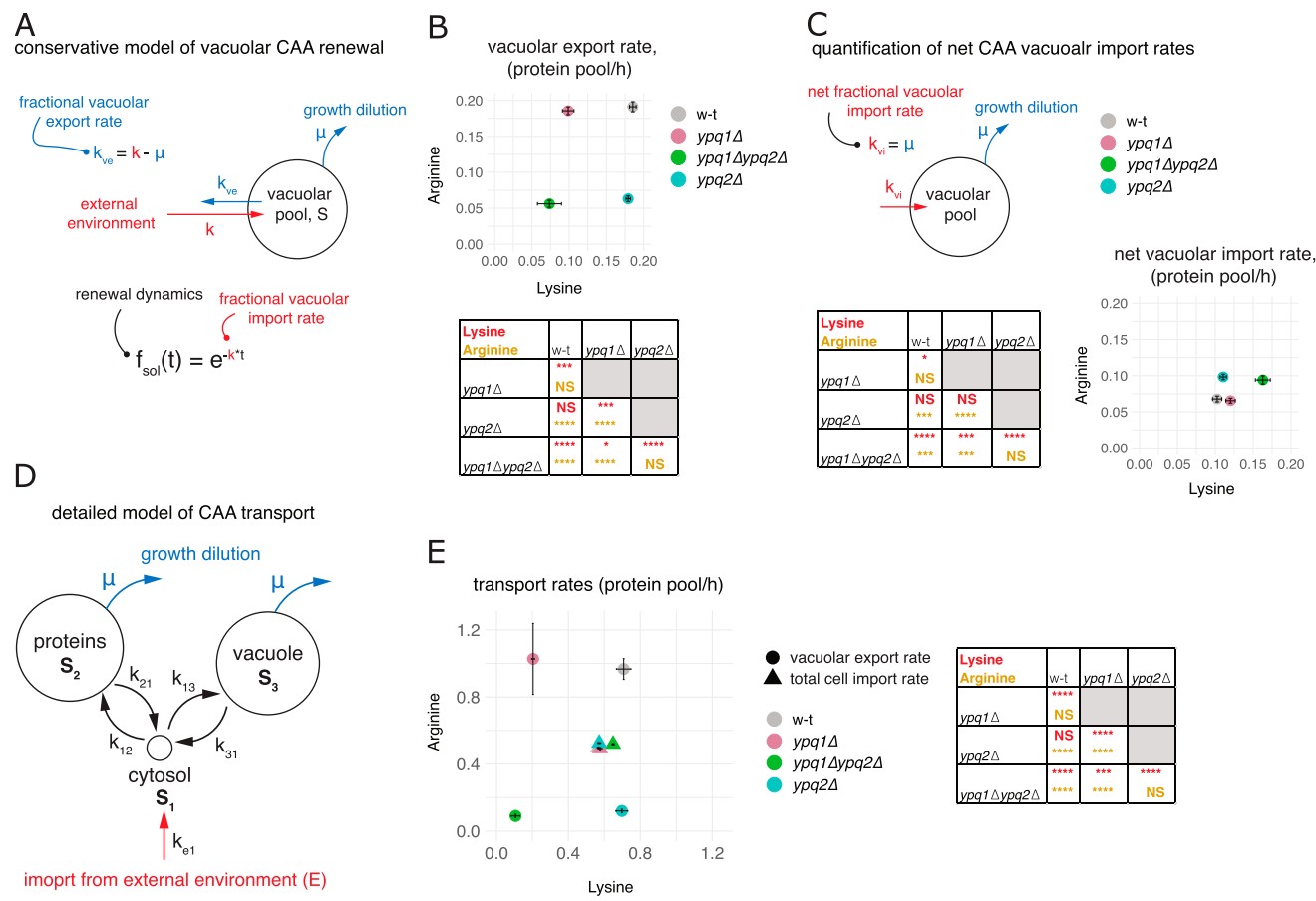

**Figure 6. Ypq1 and Ypq2 are highly efficient mediators of vacuolar CAA export.**
**(A)** Conservative model of vacuolar CAA renewal. All CAA import into the vacuole is attributed to new CAA acquired from the outside environment. In this case the fractional vacuolar import rate k (i.e., the number of vacuolar CAA pools imported per unit of time), is directly connected to the experimentally observed renewal dynamics of the vacuolar CAA, $f_{sol}(t)$, and can be quantified using these measurements (see the Materials and Methods section). Because at steady state vacuolar import is balanced by vacuolar export and the cell culture growth at rate $\mu$, the fractional vacuolar export rate, denoted $k_{ve}$ (number of vacuolar CAA pools exported per unit of time), can be further quantified as k–$\mu$ using measured cell culture growth dynamics to determine $\mu$. **(B, C, D, E)** Quantification of CAA transport rates based on the analysis of the experimental dynamic labeling readouts described in Fig 5. **(A, B)** Vacuolar export rates evaluated under the framework described in (A). The vacuolar export rates are expressed in units of the protein-borne CAA exported per hour to adjust for differences in the vacuolar pool size between genetic conditions (****: $P < 0.0001$, ***: $P < 0.001$, *: $P < 0.05$ by one-way Tukey HSD). **(C)** Top: Determination of net vacuolar import rates. Net fractional vacuolar CAA pools imported per unit of time to sustain vacuolar pool growth) is equal to cell culture growth rate $\mu$ in steady-state growth conditions and therefore can be determined directly from the culture growth measurements. Bottom: Net CAA vacuolar import rates expressed in units of the protein-borne CAA imported per hour (****: $P < 0.0001$, ***: $P < 0.001$, *: $P < 0.05$ by one-way Tukey HSD). **(D)** Detailed model of CAA transport that accounts for the fluxes between protein, vacuolar, and cytosolic pools of CAA. The parameters of the model $k_{ij}$ are the rates of CAA transfer from the source pool i to the recipient pool j, measured in units of the size of the recipient pool $S_j$. All pools are considered to grow at the biomass growth rate $\mu$. **(D, E)** Vacuolar export rates of CAA evaluated using the detailed model described in (D) based on the experimentally observed CAA renewal in the vacuolar, protein and total fractions. The rates are expressed in units of protein-borne CAA transported per hour and are shown in comparison with the total cellular CAA import rates in the same units. (****: $P < 0.0001$, ***: $P < 0.001$, **: $P < 0.01$ by one-way Tukey HSD). Error bars (SD) and $P$-values are evaluated based on n = 3 biological replicates.

We first set out to evaluate the vacuolar export rates conservatively assuming that all internal contributions to CAA import into the vacuole are neglected such that the entire vacuolar CAA influx is attributed to the new amino acids, acquired directly from the exchanged medium (Fig 6A). Under this conservative assumption, fractional vacuolar import rate can be determined directly using experimentally observed renewal dynamics and then used to evaluate the export rates as discussed above (Fig 6A; see the Materials and Methods section). The adjusted WT CAA export rates evaluated under this framework were equivalent to ~20% of the CAA content in the cellular proteins exported per hour (Fig 6B). To put these values in perspective, we compared them with net CAA vacuolar import rates necessary to support the growth of the vacuolar mass. Because vacuoles grow at the cell culture growth rate in steady-state conditions, the net vacuolar import rates can be determined simply by adjusting the cell culture growth rate for the vacuole-to-protein pool size ratios (see the Materials and Methods section). This showed that CAA export rates in WT cells must exceed net vacuolar import by two to three times (Fig 6B and C). In other words, the WT cells must keep importing CAA at least three times faster than necessary to support steady-state vacuolar growth. Furthermore, the vacuolar export rates evaluated under

this framework for lysine and arginine were selectively reduced by two to three times in deletions of *YPQ1* and *YPQ2*, respectively ($P < 0.0001$ by two-way ANOVA), while not altering or even increasing the corresponding net vacuolar import rates (Fig 6B and C). These results align well with our qualitative analysis of the roles of Ypq1 and Ypq2 in CAA transport, leading us to conclude that Ypq1 and Ypq2 mediate efficient and rather selective vacuolar export of lysine and arginine in actively growing cells.

Our conservative model will always underestimate the real CAA import rates and, hence, the export rates as it does not account, for example, for the reimport of exported amino acids back into the vacuole or import of protein-borne amino acids. Therefore, the real magnitudes of vacuolar export are likely higher. These effects could also potentially influence evaluation of differences in vacuolar export rates between mutants. To determine export rates more accurately, we developed a detailed model of CAA transport using a compartmental modeling approach (Cobelli et al, 2000; Onischenko et al, 2020; Noor et al, 2025 *Preprint*). Our detailed model included vacuolar and protein-borne amino acid pools represented as separate compartments that can acquire CAA from an external environment and exchange them with each other through a small cytosolic pool (Fig 6D). We assumed that during active growth, cells neither significantly export CAA to the outside environment nor degrade them on the time scale of our assays. These assumptions were validated by analyzing the light CAA content in the heavy CAA medium inoculated with the light CAA labeled yeast cells and by quantifying changes in the total CAA content in the whole culture (cells plus medium) during long-term cell culture growth, respectively (Fig S8A and B). The absence of significant degradation and cellular export allowed us to reduce the number of free parameters in the model and to fit them reliably using the experimental CAA labeling dynamics in the protein and vacuolar pools. This more accurate analysis showed that explaining the observed labeling dynamics in WT cells required vacuolar export rates four to five times higher than according to our conservative evaluations (Figs 6B and E, and S8C and D). This further supports the view that bulk CAA exchange between vacuoles and cytosol during active growth is much faster than the net import of CAA (Fig 6C and E). For comparison, we also determined the total cellular import rates of external lysine and arginine using the renewal dynamics of the total cellular pools of these amino acids (see the Materials and Methods section). This showed that the vacuolar export rates of both lysine and arginine are comparable with their total cellular import rates (Fig 6E). Consistent with the results obtained with the conservative model, these more accurate evaluations also showed selective fourfold–eightfold impairment of lysine and arginine export conferred by deletions of *YPQ1* and *YPQ2* (Fig 6E). Interestingly, this also revealed a noticeable decrease in lysine export rates in *ypq1Δypq2Δ* mutant compared with *ypq1Δ* cells (Fig 6E). Together with the additive effects of *YPQ2* deletion in *ypq1Δ* background reducing vacuolar lysine renewal (Fig 5E) and increasing the vacuolar lysine pool (Fig 5D), this pattern supports incomplete selectivity of Ypq2 for arginine export.

Together, we conclude that, during active cell growth, Ypq1 and Ypq2 mediate a highly efficient and mainly selective export of lysine and arginine, respectively, far exceeding the net vacuolar import flux maintaining steady-state growth of the vacuolar CAA stores. At the same time, Ypq2 may be incompletely selective for arginine export and may partially mediate export of lysine (Fig 8).

## Ypq1 mediates vacuolar import of lysine in the absence of a proton gradient

Our findings that Ypq1 and Ypq2 confer high export rates of CAA across the vacuolar membrane (Fig 6E) imply that it must be compensated by similarly high rates of vacuolar import. Because Vsb1-mediated import being dependent on V-ATPase function is likely coupled to consumption of ATP, this could result in a futile import–export cycle unless there are alternative import routes. Intriguingly, previous studies have shown that Ypq1 can also import lysine into isolated vacuolar vesicles and reconstituted proteoliposomes (Sekito et al, 2014a; Arines et al, 2024). In addition, Ypq2 has been shown to catalyze bidirectional transport of arginine into isolated vacuoles and vesicles (Kawano-Kawada et al, 2019; Cools et al, 2020). This raises the possibility that Ypq1 could also mediate lysine import into the vacuole independently of the proton gradient.

To test this hypothesis, we took advantage of the fact that the *vph1Δ* mutant, in which the V-ATPase is inactivated (Nishi et al, 2003; Li & Kane, 2009), is unable to establish a detectable intracellular lysine pool (Fig 1A). Thus, we applied the cytochrome C cell permeabilization method to analyze the distribution of $^{14}$C-lysine between cytosolic and vacuolar fractions of a WT and *vph1Δ* strain. As in previous experiments, assays were performed in *rsp5* mutant backgrounds, adjusting $^{14}$C-lysine concentrations accordingly to ensure its comparable uptake (Fig 7A). Surprisingly, the *rsp5vph1Δ* mutant accumulated $^{14}$C-lysine and its derivatives predominantly in the vacuolar fraction, like the *rsp5* mutant (Fig 7B). To determine whether this vacuolar accumulation could be attributed to Ypq1, we repeated the assay with an *rsp5vph1Δypq1Δ* triple mutant. At comparable levels of uptake (Fig 7A), in this case ~60% of $^{14}$C-lysine and its derivatives were recovered in the cytosolic fraction. Because the *rsp5vph1Δ* mutant is defective in vacuolar acidification (Tarsio et al, 2011) and fails to establish a significant vacuolar lysine pool, Ypq1-mediated uptake of lysine into the vacuole likely occurs independently of the proton gradient.

From these observations, we conclude that Ypq1 functions as a facilitator able to catalyze transport of lysine in or out of the vacuole depending on substrate electrochemical gradient.

# Discussion

In this study, we characterized Vsb1 as the main lysine importer at the yeast vacuole. Consistent with prior work (Cools et al, 2020; Kawano-Kawada et al, 2021), Vsb1 localizes to the vacuolar membrane and its loss provokes a reduction of total CAA pools and impairs long-term accumulation of exogenous lysine. Further supporting its role in lysine import, *VSB1* overexpression increases intracellular free lysine levels (Fig 1A). Given that about 90% of CAA are normally stored in the vacuole (Messenguy et al, 1980; Kitamoto

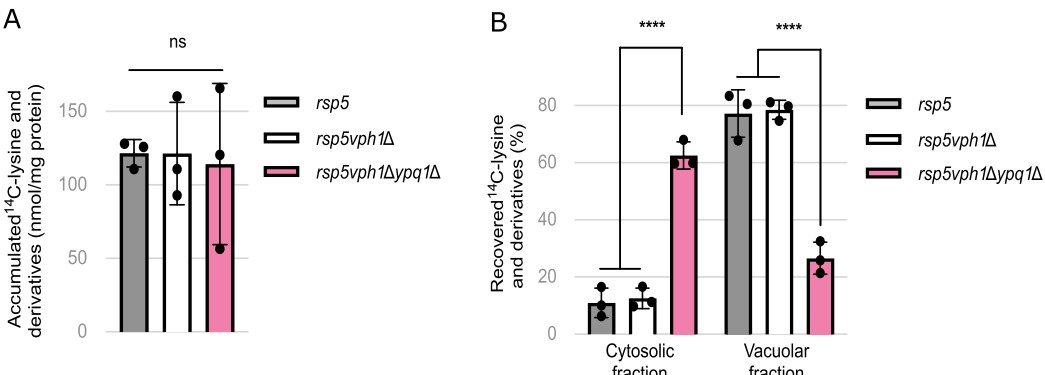

**Figure 7.  Ypq1 mediates vacuolar import of lysine in the absence of a proton gradient.**
**(A)** The ¹⁴C-lysine accumulated in the *rsp5* (30 µM ¹⁴C-lysine), *rsp5vph1Δ* (90 µM ¹⁴C-lysine) and *rsp5vph1Δypq1Δ* (90 µM ¹⁴C-lysine) strains (ns: $P > 0.05$ by one-way ANOVA test) (n = 3 biological replicates). **(A, B)** The distribution of initially accumulated ¹⁴C-lysine and derivatives (from (A)) between the cytosolic and vacuolar fractions after cell permeabilization with cytochrome C in the *rsp5*, *rsp5vph1Δ*, and *rsp5vph1Δypq1Δ* strains (****: $P < 0.0001$ by one-way ANOVA) (n = 3).

et al, 1988), their total levels serve as a reliable proxy for vacuolar pools. Nevertheless, permeabilization assays showed that, in the *vsb1Δ* mutant, exogenous lysine accumulates in the cytosol instead of the vacuole (Fig 2F).

As steady-state lysine levels were only reduced by ~50% in cells lacking Vsb1, additional vacuolar lysine importers are likely to exist. We tested candidates, including the previously described Vba1–3 importers (Shimazu et al, 2005), but deletion of their corresponding genes had no significant effect on lysine levels under our assay conditions (Fig 1A). Although we did not measure Vsb1 activity in vitro, previous work using HA-tagged Vsb1 in reconstituted vesicles showed that its arginine transport activity relies on the V-ATPase-generated proton gradient (Kawano-Kawada et al, 2021). Our data are consistent with Vsb1 being a proton antiporter, as the *vph1Δ* mutant, which lacks a functional vacuolar V-ATPase, fails to establish a lysine pool, whereas bafilomycin A treatment blocks vacuolar accumulation of exogenous lysine in WT cells (Figs 1A and 2B). However, indirect effects of the *VPH1* deletion or bafilomycin A on transporter activity cannot be ruled out.

Because the structure of Vsb1 has not yet been elucidated, we analyzed an Alphafold 3 model of Vsb1 as a dimer, considering the oligomeric state of SLC26A/SulP transporters (Fig 3A). Despite its rather poor quality, the model is sound in the light of our experiments showing the importance of the cytoplasmic domains and several key residues in the transmembrane domain. Compared with characterized eukaryotic SLC26A/SulP transporters, Vsb1 has two additional domains, an N-terminal tail of unknown fold and function and a C-terminal RmlC-like domain. Whereas truncating the N-terminal tail did not affect CAA accumulation, the absence of the RmlC-like domain led to a *vsb1Δ*-phenotype. This domain is often associated with nucleotide or metal ion binding (Dunwell et al, 2001), but such a function seems unlikely for Vsb1 because the dimerization does not restore the canonical RmlC binding site (Giraud et al, 2000). Considering the extensive interaction surface of 2,180 Å² between the RmlC-like domains in Vsb1, it could have a role in oligomerization like in the potassium channel Eag1 (Whicher & MacKinnon, 2016). Of note, the

385 Å²-surface interaction between the STAS domains in the Vsb1 model is unusually small compared with those reported for *Homo sapiens* Slc26A2 (~ 1,100 Å) (Hu et al, 2024), *Meriones unguiculatus* Slc26A5 (~ 1,200 Å) (Butan et al, 2022), and *Mus musculus* Slc26A9 (~ 1,300 Å) (Walter et al, 2019). Even though Vsb1 comprises a core membrane transporter-like domain, we cannot completely exclude the possibility that it does not function as a transporter, as seen in the cochlear outer hair cell motor prestin (Butan et al, 2022) or the yeast Ssy1 receptor of extracellular amino acid (Didion et al, 1998; Iraqui et al, 1999). In that case, Vsb1 may act as a regulator for CAA vacuolar transport. Our mutational study of the putative binding site within the transmembrane domain, however, argues in favor of Vsb1 being a transporter. Indeed, the substitution of Asp-223, Tyr-227, and Glu-278 led to a loss of CAA accumulation and an increased canavanine sensitivity (Fig 3E and F). Specifically, Asp-223 is likely an adaptation of the binding site to accommodate a cation instead of an anion, considering that a glutamine residue is found at the same position in all Slc26A anion transporters. As expected from the structures of SLC26A anion transporters, Tyr-227 could be involved in the transport of CAA via H-bond mediated interaction. Because Glu-278 is near the predicted binding site, we hypothesize that its carboxylate could neutralize the TM3 macro-dipole, which is consistent with cation binding. Glu-278 could also play a role in cation exchange. In fact, in BicA, SLC26Dg, and AtSULTR4, a carboxylate, brought by either an aspartate or a glutamate residue from the TM8, is found at a position almost equivalent to Glu-278 (Fig S2D) (Geertsma et al, 2015; Wang et al, 2019, 2021) and has been shown to be important for either Na⁺/anion or H⁺ exchange (Wang et al, 2019, 2021).

In addition to elucidating the role of Vsb1 in lysine import, our study provides evidence that the PQ-loop transporter Ypq1 exports lysine out of the vacuole. This conclusion is supported by the elevated lysine pool of the *ypq1Δ* mutant and its strong induction of *LYS9* upon lysine withdrawal (Fig 4A). Moreover, under lysine starvation conditions, Ypq1 is critical for the cell to mobilize vacuolar lysine previously accumulated via Vsb1, and to sustain residual growth (Fig 4C and E). Interestingly, the deletion of *YPQ1* still allows ~30% of the starting lysine pool to be exported,

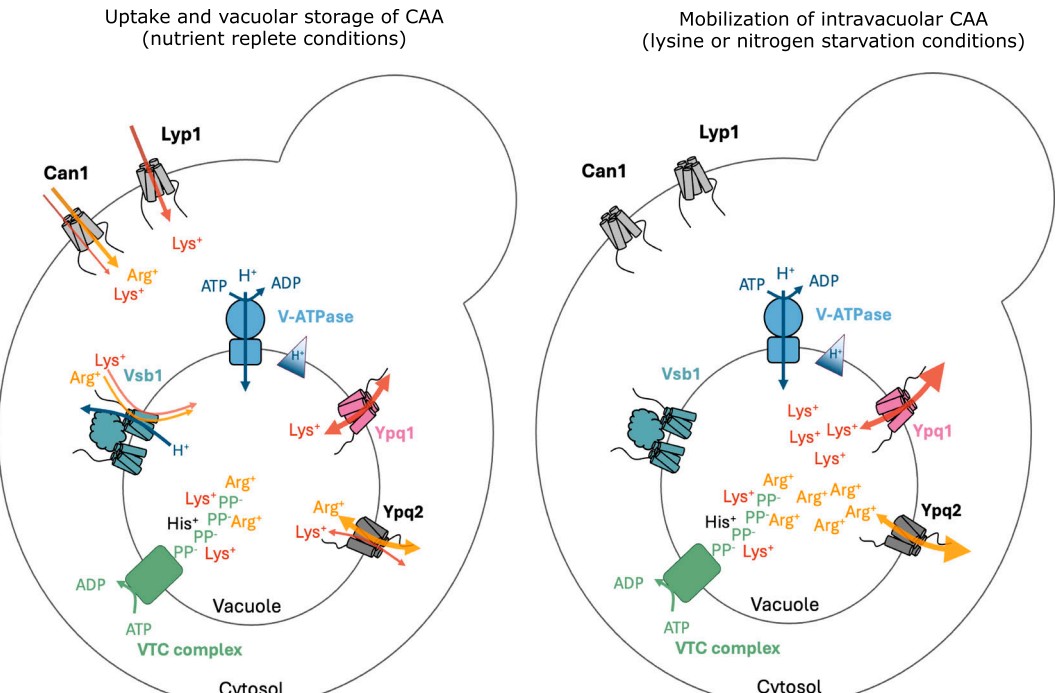

**Figure 8. Model for the dynamic storage of CAA in the vacuole.**
Under nutrient replete conditions, lysine and arginine are transported into the cytosol from the external environment by the plasma membrane transporters Lyp1 and Can1 and subsequently accumulated in the vacuole by Vsb1, whose activity is likely dependent on the proton gradient established by the V-ATPase. At the vacuolar membrane, Ypq1 and Ypq2 act as facilitators, able to catalyze bidirectional transport. Ypq1 primarily transports lysine, whereas Ypq2 transports arginine and, to a lesser extent, lysine. Lysine and arginine are retained in the vacuole through electrostatic interactions with the negatively charged polyphosphate chains synthesized by the VTC complex. Under lysine starvation, Ypq1 mediates net export of lysine, and under nitrogen starvation, arginine is mobilized to the cytosol by Ypq2. Vsb1 is presumably inactive under starvation conditions. The export of CAA sustains cell growth under nutrient limitation.

indicating the involvement of additional exporters. We have ruled out the proposed vacuolar CAA transporters Ypq2, Ypq3, and Avt4 as contributors to lysine export under lysine starvation (Fig 4D). This implies that other vacuolar lysine exporters remain to be discovered.

Furthermore, we confirm that Ypq1 undergoes down-regulation under lysine starvation, albeit to a lesser extent than previously reported (Fig S5B) (Li et al, 2015a, 2015b; Zhu et al, 2017; Arines et al, 2021). Although this process has been characterized under the assumption that Ypq1 functions as a lysine importer, down-regulation was proposed to occur when lysine depletion in the cytosol stabilizes a Ypq1 conformation recognized by the adaptor Ssh4, which recruits Rsp5 to ubiquitinate Ypq1 and trigger ESCRT-dependent vacuolar degradation (Li et al, 2015b; Arines et al, 2021). Importantly, Ypq1 down-regulation in the absence of lysine does not contradict a role in lysine export. Our measurements show that complete Ypq1 degradation occurs several hours after lysine withdrawal, when vacuolar lysine stores are nearly exhausted (Fig 4C) (Li et al, 2015b). We, therefore, propose that Ypq1 primarily functions as an exporter and adopts the Ssh4-recognized conformation in response to declining vacuolar, rather than cytosolic, lysine levels. Consistent with this model, purified Ypq1 is destabilized in proteoliposomes lacking luminal lysine (Arines et al, 2024).

Our dynamic labeling experiments using stable isotope coded amino acids revealed that, not only in starvation conditions but also during active growth, cells continuously export CAA from the vacuole (Fig 5C). Kinetic modeling and exploration of the effects of *ypq1Δ* and *ypq2Δ* deletions indicate that Ypq1 and Ypq2 are the key but incompletely selective mediators of this export (Fig 5). Surprisingly our quantitative analysis of CAA transport indicates that CAA export rates are comparable with the total cellular import rates of lysine and arginine and exceed the net vacuolar import that supports accumulation of these amino acids in the growing vacuoles roughly by an order of magnitude (Fig 6C and E). This raises the intriguing question of how cells maintain even higher vacuolar import to accomplish the CAA accumulation. Should the import of lysine and arginine be solely driven by Vsb1 and other vacuolar proton antiporters, the cell would incur a significant energetic cost just to maintain vacuolar CAA levels. As an alternative, Ypq1 and Ypq2 may function as facilitators (Carruthers, 1990; Leray et al, 2021), catalyzing bi-directional transport of lysine and arginine (Fig 8) whose net concentration could be retained in the vacuole at least partially through electrostatic interactions, for example, with polyphosphate chains (Dürr et al, 1979). Supporting this hypothesis, our current study demonstrates that Ypq1 can mediate lysine import in the absence of a proton gradient across the vacuolar membrane (Fig 6B) and we previously observed that Ypq2 can mediate apparent exchange of arginine in isolated vacuoles (Cools et al, 2020). However, a more detailed in vitro analysis would be needed to unequivocally determine the transport mechanisms of Ypq1/2 and to understand how environmental conditions and electrochemical gradients modulate their activity.

Finally, our findings illustrate how the vacuole can serve as a dynamic reservoir that regulates CAA homeostasis. In fact, the plurality of CAA transporters at the vacuolar membrane can be explained as a way for a unicellular organism such as yeast to respond quickly to variations in CAA availability. The putative proton antiporter Vsb1 imports lysine and arginine into the vacuole under replete conditions. This transport into the vacuole is key to promote high levels of CAA uptake through the plasma membrane permeases, Lyp1 and Can1 (Fig 2B) (Shimazu et al, 2005; Cools et al, 2020). Although energetically costly for the cell as it requires a secondary active transporter, lysine storage mitigates its toxicity, especially under poor nitrogen conditions, and stored CAA pools can be mobilized later. In addition, the Ypq1/2 facilitators are constitutively active in growing cells and dampen fluctuating cytosolic CAA concentrations, preventing excessive biosynthesis under replete conditions (Figs 1A and 4A) and promoting quick release to sustain growth under nutrient limitation.

Beyond understanding vacuolar CAA homeostasis, our results may have important practical implications. For example, protein stabilities and degradation kinetics are often analyzed in various model systems including yeast by dynamic SILAC assays, which typically use stable CAA isotopologues as protein labels (Christiano et al, 2014; Freeman Rosenzweig et al, 2017). Since actively growing yeast cells appear to have large and dynamic vacuolar CAA stores that significantly influence the labeling dynamics of cellular proteins in a manner dependent on YPQ1 and YPQ2 (Fig 5E), these effects would lead to delayed labelling of proteins and must be taken into account to correctly interpret the results of such experiments (Noor et al, 2025 Preprint).

# Materials and Methods

### Yeast strains, plasmids, and growth conditions

The yeast strains used in this study (Table S2) were derived from the reference WT Σ1278b (Bechet et al, 1970), except for dynamic labeling experiments (see below). All knockout strains as well the GFP-tagged YPQ1 strains were produced using the gene disruption cassette integration method (Guldener, 1996) with the oligonucleotides of Table S3, as previously reported (Goldstein & McCusker, 1999; Lee et al, 2013). Strains were verified by PCR with primers specific to the ORF, the promoter and the terminator of the respective genes and the selection marker. Except where otherwise stated, cells were grown at 29°C on a minimal buffered medium (pH = 6.1) (Grenson & Acheroy, 1982) with glucose (3%) as the carbon source and ammonium in the form of $(NH_4)_2SO_4$ (10 mM) as the nitrogen source. To complement the uracil auxotrophy, either uracil was added at 0.0025% or strains were transformed with the pFL38 (Bonneaud et al, 1991) plasmid. When used, lysine was supplemented at a final concentration of 500 $\mu$M unless stated otherwise. The final concentrations of substances added to solid or liquid media were for canavanine 0.6 $\mu$g/ml and for bafilomycin A 9 $\mu$M. In all experiments, cells were examined or harvested during exponential growth, a consistent number of generations after seeding. The plasmids used in this study (Table S4) were constructed using in vivo homologous recombination in

yeast, as previously described (Merhi et al, 2011; Gournas et al, 2017). Plasmids were cloned by rescuing in Escherichia coli and verified by sequencing. The sequences of the oligonucleotides are available upon request.

### Measurement of total soluble lysine pools

Yeast cultures (25 ml) were collected in exponential phase (~0.4 × $10^7$ cells/ml) by centrifugation (7,000$g$ for 3 min) and washed three times with 20 ml ultrapure water. Cells were resuspended in 2 ml ultrapure water and boiled for 15 min. The extract was centrifuged (13,000$g$ for 3 min) to collect condensation drops and the supernatant was filtered through a PVDF hydrophilic syringe filters (0.2 $\mu$m, ROCC) to remove cell debris. Amino acid content was quantified using the AccQ-Tag Ultra turn-key Method (Waters), following the manufacturer's protocol. Another 25 ml of the same culture was filtrated through an MCE membrane filter (0.45 $\mu$m, ROCC) and dried at 60°C for 24 h to measure dry cell weight for normalization. Data are reported as mean values, with error bars representing SD.

### Uptake assays in whole cells

Accumulation of $^{14}$C-labeled lysine (Perkin-Elmer) in whole cells was measured at the indicated time points in whole-cell uptake assays, as described previously (Merhi et al, 2011; Ghaddar et al, 2014; Gournas et al, 2017). Initial uptake rates were determined by sampling accumulated counts (cpms) at 15 s intervals within 1 min of uptake, where the kinetic of $^{14}$C-lysine accumulation is linear. All measurements are expressed in nmol/mg protein per unit of time and reported as mean values, with error bars representing SD.

### Cell permeabilization assays by cytochrome C

Plasma membrane permeabilization by cytochrome C was performed as described previously (Cools et al, 2019). After a 10-min whole-cell uptake assay, two 5 ml aliquots of culture were filtered onto an MCE membrane filters (0.45 $\mu$m, ROCC) and washed with ultrapure water. To measure the total counts per minutes (cpm) of the initial culture, one membrane containing cells was placed in 3 ml of scintillation fluid (Ultima-Flo AP) and radioactivity was measured using a Beckman Coulter LS 6500 Liquid Scintillation Counter (Beckman Coulter). For plasma membrane permeabilization, the second membrane was incubated in 4 ml of cytochrome C solution (1 mg/ml in 1 M sorbitol) for 1 h at 4°C with gentle shaking. The cell suspension was then percolated over a glass microfiber filter (GF/C 25 mm) and washed three times with 1 ml of 1 M sorbitol. The combined flow-through and wash fractions were collected as the "cytosolic fraction." To release vacuolar contents, 2 × 3 ml of distilled water was applied to the filter, and the resulting flow-through was collected as the "vacuolar fraction." For scintillation counting, 500 $\mu$l of the cytosolic fraction was mixed with 18 ml of scintillation fluid, and 1 ml of the vacuolar fraction was combined with 6 ml of scintillation fluid. Cpm values of the cytosolic and vacuolar fractions were normalized to the total cpm

of the initial culture and reported as mean values, with error bars representing SD.

## Fluorescence microscopy

Cells in exponential phase (~2 × 10^6 cells/ml) were laid down on a thin layer of 1% agarose. They were viewed at room temperature with an epifluorescence microscope (Eclipse Ci-L; Nikon) equipped with a 100x differential interference contrast N.A. 1.40 Plan Apochromat objective, and appropriate filters. Images were captured with a digital camera (IMAGONGSOURCE TV Lens C-0.45x, Nikon) and NIS-Element D acquisition software (Nikon) and were processed with Fiji software (Schindelin et al, 2012). In each figure, we typically show only a few cells, representative of the whole population. Labeling of the vacuolar lumen with CMAC (7-amino-4-chloromethylcoumarin; Thermo Fisher Scientific) was performed by adding the fluorescent dye to a concentration of 25 µM at least 30 min before visualization. Labeling of the vacuolar membrane of whole cells with FM4-64 (N-(3-Triethylammoniumpropyl)-4-(6-(4-(Diethylamino) Phenyl) Hexatrienyl) Pyridinium Dibromide; Thermo Fisher Scientific) was performed as described previously (Vida & Emr, 1995). For quantifications, images were analyzed with custom-made FIJI macros, calculating the vacuolar membrane-to-total vacuolar intensity. Briefly, for membrane-to-total fluorescence intensity, two homocentric ellipses outlining the whole vacuole or the whole vacuole excluding the vacuolar membrane were manually drawn in middle-section images. The intensities of the channels of interest were measured within manually selected cell outlines, whereas the median of the fluorescence intensity in the whole image was subtracted as background. All parameters were calculated from at least two independent biological replicates for each condition. The values for single vacuoles are presented in violin plots. After verification that two independent biological replicates gave statistically nonsignificant differences in mean values, the values of the two experiments were merged.

## Growth curves

Comparative analyses of growth in different conditions were performed by growing cells in a 24- or 96-well non-treated microplate (VWR) incubated at 30°C with fast shaking (600 rpm) into a SPECTROstar Nano microplate reader (BMG Labtech). Cell growth was monitored by measuring the absorbance at 660 nm every 20 min for 48 h.

## AlphaFold prediction and model analysis

The Vsb1 model was predicted using AlphaFold 3 server (Abramson et al, 2024). Protein structure comparisons were performed using DALI server (Holm et al, 2023). Interaction surfaces were calculated using PDBe PISA (Krissinel & Henrick, 2007). Structures were illustrated using the PyMOL molecular-graphics system version 2.5.0 (Schrödinger). The structure-based sequence alignment of Vsb1 with characterized members of the SLC26A/SulP superfamily was performed using Expresso (Notredame et al, 2000; Armougom et al, 2006) and visualized using ESPript 3.2 (Gouet, 2003).

## Western blotting

Total cell protein extracts were prepared and analyzed by SDS–PAGE as described previously (Hein et al, 1995). Proteins were transferred to a nitrocellulose membrane (Amersham Protran Premium 0.45 µm) and probed with a mouse monoclonal anti-GFP (RRID: AB_390913; Roche Applied Science), or anti-Pgk1 (PGK1 Monoclonal Antibody 22C5D8; Thermo Fisher Scientific). Primary antibodies were detected by enhanced chemiluminescence (SuperSignal West Femto Maximum Sensitivity Substrate; Thermo Fisher Scientific, or Immobilon Classico Western HRP Substrate; Merck Millipore) after treatment with horseradish-peroxidase-conjugated anti-mouse or anti-rabbit immunoglobulin (Ig) G secondary antibody (Merck Millipore). Signals were detected with an Imager CHEMI Premium (VWR).

Relative semi-quantitative amounts of total proteins were estimated from one to three biological replicates of non-saturated exposures using the gel analyzer tool of FIJI. Each band was selected by using rectangular ROI (Region Of Interest) selection and « Gels » analyzer, followed by quantification of the peak area of obtained histograms. Data were acquired as area values. In each graph, the ratios of GFP/Pgk1 normalized to the ratio of experiment which is set as 1 are plotted in a bar chart, normalized to the ratio of EXP which is set as 1.

## Quantitative RT–PCR

RNA isolation and cDNA synthesis were performed with minor modifications to the protocol described by Schmitt et al (1990). Briefly, total RNA was extracted from 4 ml exponential-phase cultures using the hot acidic phenol method (Schmitt et al, 1990). Complementary DNA (cDNA) was synthesized from 100 to 500 ng of total RNA using the RevertAid H Minus First Strand cDNA Synthesis Kit (Thermo Fisher Scientific) following the manufacturer's instructions. Purified cDNA was subsequently quantified by quantitative RT–PCR using gene-specific primers (Table S5) and the Power Track SYBR Green Master Mix (Thermo Fisher Scientific) on a LightCycler96 system (Roche Applied Science). Standard curves for each primer pair were generated using five successive 10-fold dilutions of genomic DNA. These curves were used to assess PCR efficiency and calculate the relative concentrations of target DNA in all other samples. The specificity of the PCR products was assessed by melting curve analysis. Gene expression levels were normalized to *TBP1* mRNA levels and are presented as mean values. Error bars indicate SD from replicate measurements.

## Copper chloride permeabilization assay

Yeast strains were cultivated at 30°C in a minimal buffered medium (pH = 6.1) with 3% glucose as the carbon source and 10 mM $(NH_4)_2SO_4$ as the nitrogen source. For strains derived from BY4742 *(MAT(alpha) his3Δ1 leu2Δ0 lys2Δ0 ura3Δ0)*, the medium was further supplemented with 0.006% L-leucine, 0.002% L-histidine, 0.002% uracil, 0.0025% L-lysine, and 0.002% L-arginine. When cultures reached an absorbance of 0.25 at 660 nm (~2.5 × 10^7 cells/ml), two 12.5 ml aliquots were filtered through an MCE membrane filter (0.45 µm; ROCC) and washed with ultrapure water. The

membrane was washed twice with buffer B (2.5 mM potassium phosphate pH = 6, 0.6 M sorbitol), and cells were resuspended in 2 ml of buffer A (10 mM glucose and 0.2 mM $CuCl_2$ in buffer B). After incubation at 30 °C for 5 min with gentle shaking, the cells were filtered (MCE membrane filters, 0.45 $\mu$m, ROCC) and the flow-through (fraction B) was collected for further analysis. The cells were then resuspended in 2 ml of ultrapure water, and boiled for 15 min. The boiled cell suspension was centrifuged (13,000$g$ for 3 min) to collect condensation drops and the supernatant was filtered through a PVDF hydrophilic syringe filter (0.22 $\mu$m; ROCC) to remove cell debris. The final extract (fraction C), representing the vacuolar fraction, and cytosolically enriched fraction B were subjected to amino acid analysis by mass spectrometry.

The vacuolar lysine and arginine content in the total soluble fraction was determined using glutamate or phenylalanine as a fiducial cytosolic marker (Kitamoto et al, 1988) and assuming that the exposure to $CuCl_2$ does not significantly release the vacuolar content. In this case, the vacuolar fraction in the total soluble pool, $f_{vac}$, can be quantified as:

$$f_{vac} = 1 - \frac{R_b}{R_c}$$

where $R_b$ and $R_c$ are relative amino acid enrichments determined as the signal intensity ratio of the given amino acid and phenylalanine in fractions B and C, respectively. Because some vacuoles can be broken by $CuCl_2$ treatment, this provides a conservative evaluation of the size of the vacuolar fraction.

## Metabolic labeling assays

For dynamic labeling assays yeast strains derived from BY4742 *(MAT(alpha) his3Δ1 leu2Δ0 lys2Δ0 ura3Δ0)* were cultivated at 30°C in a standard synthetic complete medium (pH = 6.1) with yeast nitrogen base containing $(NH_4)_2SO_4$ as nitrogen source and 2% glucose as carbon source. The medium was further supplemented with 0.002% each adenine, L-arginine, L-histidine, L-methionine, L-tryptophan, uracil, 0.006% L-leucine, 0.0025% L-lysine, 0.005% L-phenylalanine, 0.02% L-threonine, 0.003% L-tyrosine and designated as the "light medium." When cultures reached an absorbance of ~0.15 at 600 nm, a 10 ml sample was harvested and stored for further fractionation and amino acid extraction (see below), whereas 90 ml of the culture was harvested by filtration (WCN membrane filters, 0.8 $\mu$m; Whatman), washed on the filter with 10 ml and transferred to 90 ml of a pre-warmed heavy isotope labeled medium containing 0.0025% [$^{13}C_6$, $^{15}N_2$]-L-lysine and 0.002% [$^{13}C_6$, $^{15}N_4$]-L-arginine instead of light isotopologues. Immediately after transfer, the initial culture turbidly was measured at 600 nm. Culture samples were collected at 20, 60 and 120 min following the medium exchange where 1 ml culture volume was used to determine turbidity and 10 ml was harvested and stored at –20°C. For harvesting, the culture samples were immediately spun down for 1 min at 3,500$g$. The cell pellet was quickly transferred to a 1.5 ml tube, resuspended in 1 ml of MS grade water and spun down again for about 10 s to collect and discard the liquid fraction. The cell pellet was then immediately frozen at –20°C.

For fractionation and amino acid extraction, the pellets were resuspended in 150 $\mu$l of MS grade water each and boiled for 15 min

at 100°C. 100 $\mu$l of boiled cell suspension was spun down for 1 min at 17,000$g$ saving the supernatant, whereas the pellet was washed once with 100 $\mu$l of MS grade water followed by completely removing the liquid fraction. The remaining 50 $\mu$l of the boiled cells, and the washed pellet, representing the total and protein fractions, respectively, were dried in a speedvac for 2 h at 45°C, whereas the supernatant, representing soluble fraction, was stored at –20°C for further amino acid analysis.

The protein content of the total and insoluble fractions was hydrolyzed using 6 M hydrochloric acid (Sauer et al, 1996) as follows: The dry material was resuspended in 250 $\mu$l of freshly prepared 6 M HCl each, transferred to safe-lock microtubes, and incubated for 24 h at 110°C in a heating block. The liquid phase was then evaporated at 95°C with constant air flow and the remaining material was resuspended in 150 $\mu$l of MS grade water. After pelleting the debris for 2 min at 17,000$g$, 50 $\mu$l of supernatant was saved and stored at –20°C for amino acid analysis by quantitative mass spectrometry.

The fractional content of the light CAA in each fraction (referred to as *labeling or renewal*) was determined based on the ratio of integrated signal peak intensities of the light ($I_L$) and heavy ($I_H$) isotopologues (see Fig S6B for representative chromatographic curves) as:

$$f = \frac{W_L I_L}{W_L I_L + W_H I_H} + \frac{1}{1 + (W_H I_H)/(W_L I_L)}$$

Here, $w_L$ and $w_H$ are the relative signal response factors of light and heavy amino acid isotopologues. To determine signal response factors and to ensure that heavy/light signal ratios remined stable over a wide range of signal intensities covering our assays, before and after each analysis series, six-point dilutions of equimolar [$^{12}C_6$, $^{14}N_2$]/[$^{13}C_6$, $^{15}N_2$]-L-lysine, and [$^{12}C_6$, $^{14}N_4$]/[$^{13}C_6$, $^{15}N_4$]-L-arginine prepared in MS-grade water were injected and analysed (0.0005, 0.002, 0.008, 0.03, 0.13, and 0.5 mM) (Fig S6A and Supplemental Data 1, calibration standards). The relative signal response factors were calculated based on the mean Light/Heavy intensity ratios among the equimolar dilution series $I^*_L/I^*_H$ as:

$$w_H = \frac{1}{1 + I^*_L/I^*_H} \; ; \; w_L = \frac{1}{1 + I^*_H/I^*_L}$$

For cellular amino acid export tests 100 ml of FV1633 cell culture (*arg4Δ* mutant) grown to $OD_{600}$ 0.2 in light SCD medium was harvested, washed with the "heavy" lysine/arginine medium and inoculated in 20 ml of "heavy" lysine/arginine medium exactly as described for dynamic labeling assays. 1 ml cell culture samples collated over 1.5-h time course were fractionated by centrifugation into soluble (medium) and pellet (cell) fractions. The cell faction of each sample was dried in a speedvac and hydrolyzed with 6 M HCl as described for dynamic labeling, and the extracted amino acids were recovered in 600 $\mu$l of MS grade water. Relative amounts of lysine and arginine isotopologues in each fraction (cells and the medium) were determined for all time points using identical 1 $\mu$l injection volumes and integrated signal peak intensities of the amino acid signal as readouts.

For amino acid degradation tests 250 $\mu$l samples of FV1633 cell culture grown in light SCD medium were periodically collected

**Life Science Alliance**

starting from $OD_{600} \approx 0.1$ over a 20-h time course. The samples were boiled for 15 min at 100°C, dried in a speedvac and hydrolyzed with 6 M HCl as described for dynamic labelling assays. After hydrolysis, the amino acid samples were recovered in 250 $\mu$l of MS grade water and spiked with an identical volume of heavy lysine/arginine standard mix. The relative total amino acid content in the cell culture samples was then determined as described for the dynamic labeling assays.

### Mass spectrometry analysis

Accurate mass UPLC-HRMS analysis of the samples was performed using a Dionex UltiMate 3000 liquid chromatography system (UPLC) coupled with a Q-Exactive mass spectrometer (HRMS) and interconnected with a heated electrospray ionization source (H-ESI) (Thermo Fisher Scientific).

Separation of the analytes was performed on an Acquity Premier BEH Amide VanGuard Fit Column (1.7 $\mu$m, 2.1 mm X 100 mm; approximate column volume of 230 $\mu$l; Waters).

For analysis, the column was kept at 50°C, the injection volume was 1 $\mu$l and the flow rate was 0.2 ml min$^{-1}$ for a total run time of 15 min. Mobile phase A consisted of 3% acetonitrile and mobile phase B of 90% acetonitrile, both buffered with 10 mM ammonium acetate. Separation of individual compounds was achieved using a multistep gradient of A and B where B composition started with 85% and reduced to 80% in 2 min and further decreased to 50% over 8 min before being brought to 40% over 1 min for washout. For equilibration, the B concentration was turned back to 85% over 4 min. This equilibration time was used to ensure stable baseline conditions and reproducible retention times. In addition, the injection needle was washed for 2 min with 50% methanol, whereas 85% B was continuously flushed onto the column (~5 column volumes in total). The LC gradient was optimized to ensure good separation between several amino acids including L-lysine, heavy L-lysine, L-arginine, heavy L-arginine, L-glutamate, L-histidine, L-threonine, L-tyrosine, L-methionine, L-tryptophan, L-phenylaniline, and L-leucine (see Fig S9A).

Ions were monitored in positive targeted single ion monitoring (t-SIM) mode with a resolution of 70,000 at $m/z$ = 200 and an isolation window of 17 $m/z$, using an inclusion parameter list determined using a water-based standard mixture containing 0.5 mM of each amino acid used in this study (see the Supplemental Data 2). t-SIM is an MS1 level acquisition mode that does not involve precursor fragmentation, and a broader isolation window does not introduce MS/MS co-isolation interference. The 17 m/z window was intentionally used for the Orbitrap t-SIM scans to simultaneously monitor both the light and heavy isotope-labeled lysine and arginine, and other relevant compounds with similar mass. The high resolving power (70,000) of Orbitrap ensured clean resolution of all relevant amino acids and the targeted isotopologues in yeast extracts (see Fig S9B).

Other MS parameters were spray voltage 3.5 kV, sheath gas flow rates 48 U, auxiliary gas flow rate 11 U, sweep gas flow rates 2 U, capillary temperature 256°C, auxiliary gas heater temperature 413°C, stacked-ring ion guide (S-lens) radio frequency (RF) level 30, automatic gain control (AGC) 2 × 10$^5$ ions, and maximum injection time 200 ms.

Exact mass acquisition and quantification were carried out using the Thermo XCalibur Quan Browser 4.0.27.42 software, with 10 ppm mass tolerance. Freestyle 1.5 software from Thermo was used to extract chromatographic data points.

### Determination of transport rates and sizes of amino acid pools

In steady-state conditions, vacuolar import of amino acids supports both their continuous export to the cytosol and the growth of the vacuolar mass at the rate of biomass growth (see Fig 6A), so these rates (expressed in units of the size of the vacuolar pool) can be related as:

$$k_{ve} = k - \mu, \tag{1}$$

where $k_{ve}$ and $k$ are the vacuolar export rate and the bulk vacuolar import rate, respectively, and $\mu$ is the biomass growth rate.

For conservative evaluation of vacuolar export rates (conservative model of vacuolar CAA renewal; Fig 6A), all heavy amino acids imported into the vacuole were considered to originate directly from the outside environment. In this case, labeling of the vacuolar pool (the light fraction) after the medium exchange, $f_{sol}(t)$, can be described by a single-exponential decay function as:

$$f_{sol}(t) = e^{-kt}, \tag{2}$$

where k is the bulk vacuolar import rate.

By taking the natural logarithm of both sides, Equation (2) can be converted to a linear relationship:

$$-\ln f_{sol}(t) = kt, \tag{3}$$

Based on this, bulk vacuolar import rates were evaluated as the slope of linear regression using negative logarithmically transformed labeling values to parametrize the linear model.

The growth rates, $\mu$, were determined analogously considering that the turbidity and the growth rate in exponentially growing cultures are related as:

$$\ln \frac{OD_{600}(t)}{OD_{600}(0)} = \mu t$$

After quantifying the bulk import rates, k, and the growth rates, $\mu$, the vacuolar export rates, $k_{ve}$ were determined using Equation (1).

The net vacuolar import rates (expressed in units of the size of the vacuolar pool) $k_{vi}$ were determined directly from biomass growth rate as:

$$k_{vi} = \mu$$

considering that during balanced exponential growth the net vacuolar import supports the net growth of the vacuolar pool at the biomass growth rate (Fig 6C).

The total cellular import rates of amino acids $k_{tot}$ were determined in the same way as described for the bulk vacuolar import rates k (see Equation (3)), that is, considering that all heavy amino acids are imported into the cell directly from outside, and that during exponential growth the labeling of the total cellular CAA pool $f_{tot}(t)$ is related to $k_{tot}$ as:

$$-\ln f_{tot}(t) = k_{tot}t$$

Relative weights of the protein and vacuolar pools of CAA, $S_{prot}$, and $S_{vac}$ were determined based on the CAA labeling in the total, soluble and protein pool at each time point using $f_{tot}(t)$, $f_{sol}(t)$, and $f_{prot}(t)$, respectively, fractional content relation and considering that all cellular CAA are partitioned between the protein and vacuolar pools:

$$f_{tot}(t)S_{tot}(t) = f_{sol}(t)S_{vac}(t) + f_{prot}(t)S_{prot}(t)$$

Because the pool weights add up to 1, that is,

$$S_{vac}(t) + S_{prot}(t) \quad S_{tot}(t) \quad 1$$

$S_{prot}$ and $S_{vac}$ can be quantified as:

$$S_{prot}(t) = \frac{f_{tot}(t) - f_{sol}(t)}{f_{prot}(t) - f_{sol}(t)}, \qquad (4)$$

$$S_{vac}(t) = \frac{f_{prot}(t) - f_{tot}(t)}{f_{prot}(t) - f_{sol}(t)}. \qquad (5)$$

The ratio $S_{vac}(t)/S_{prot}(t)$ referred to as the *vacuole-to-protein pool ratio* was quantified at each labeling time point to ensure balanced growth and for comparison of relative CAA vacuolar pool sizes between mutants (see Fig 5D).

The consensus vacuole-to-protein pool ratios were quantified for each culture based on $S_{prot}$ and $S_{vac}$ values averaged across all three labeling time points and were used to express the vacuolar transport rates in units of CAA pool contained in cellular proteins:

$$K_{ve} = \frac{k_{ve}S_{vac}}{S_{prot}}$$

$$K_{vi} = \frac{\mu S_{vac}}{S_{prot}}$$

Likewise, the averaged weights of the protein pool $S_{prot}$ were used to convert total cellular import rates to units of CAA pool contained in cellular proteins:

$$K_{tot} = \frac{k_{tot}}{S_{prot}}$$

### Construction and parametrization of the detailed model of CAA transport

The detailed model of CAA transport was constructed using a compartmental modelling (CM) approach (Cobelli et al, 2000; Onischenko et al, 2020) and parameterized based on the experimental dynamic labelling readouts using an open-source Symbolic Compartmental Models package available at https://gitlab.com/elad.noor/symbolic-compartmental-model (Noor et al, 2025 *Preprint*). In brief, CAA pools, including cytosolic, vacuolar, and protein-borne CAA, were represented as well-mixed compartments that exchange material with each other. In CMs the pools are assigned weights $S_i$ which together add up to 1. The transfer of material between pools is described by contributed turnover parameters $k_{ij}$, where the indices i and j represent the source and recipient pool, respectively, and are measured in units of the size of the recipient pool. During steady-state growth, the weights of the pools and the contributed turnovers remain constant.

The graphical representation of our CAA transport model is detailed in the Fig 6D. Specifically, the cytosolic pool $S_1$ receives CAA from external environment E at a rate $\kappa_{e1}$ and exchanges them with the protein pool $S_2$ and the vacuolar pool $S_3$. The contributed turnovers $\kappa_{13}$ and $\kappa_{31}$ define the rates of vacuolar import and export, respectively, whereas $\kappa_{12}$ and $\kappa_{21}$ define the rates of CAA use for protein biosynthesis and their backflux to the cytosolic pool after protein degradation. The blue arrows represent growth dilution of CAA pools.

In steady-state dynamic labeling experiments, the dynamics of the unlabeled fraction in compartments is described through linear inhomogeneous ordinary differential equations that have an analytical solution as the sum of exponential decay functions (Noor et al, 2025 *Preprint*). Specifically, in the case of our dynamic labeling setup in which unlabeled metabolites are fully replaced with the labeled analogue in growth medium, the dynamics of the unlabeled (light) fraction f (t) in each compartment, which we refer to as *labeling*, can be described through matrix exponentiation in the following form:

$$\boldsymbol{f}(t) = e^{\boldsymbol{M}t}\boldsymbol{1}_n$$

where $\boldsymbol{f}(t)$ is a vector of labeling values in each compartment at the time point t after the medium exchange, $\boldsymbol{1}_n$ is a unit vector, and $\boldsymbol{M}$ is a transition matrix that describes amino acid transfer rates between compartments.

In our model $\boldsymbol{f}(t)$ has 3 components that define the labeling of pools $S_1$, $S_2$, and $S_3$ and the matrix $\boldsymbol{M}$ has the following form:

$$\boldsymbol{M} = \begin{pmatrix} -k_{e1} - k_{21} - k_{31} & k_{21} & k_{31} \\ k_{12} & -k_{12} & 0 \\ k_{13} & 0 & -k_{13} \end{pmatrix}$$

The parameters of the model can be related to a set of measurable and quantifiable parameters by the constraints of mass balance (meaning that the total influx and efflux of each pool must match).

That the influx from external environment supports the growth of the whole system at the cell culture growth rate $\mu$ can be expressed as:

$$k_{e1}S_1 = \mu(S_1 + S_2 + S_3)$$

Considering that all pool weights add up to 1, we can express $\kappa_{e1}$ through this constraint as:

$$k_{e1} = \frac{\mu}{S_1}$$

Defining the efflux from the protein pool $S_2$ in units of protein pool size, $\kappa_{pd}$, as a quantifiable parameter that describes the protein degradation rate and defining $\kappa_{ve}$ as the fractional vacuolar export rate and using the above conventions we, in addition, get:

$$k_{21} = \frac{k_{pd}S_2}{S_1}$$

$$k_{12} = \mu + k_{pd}$$

$$k_{31} = \frac{k_{ve}S_3}{S_1}$$

$$k_{13} = \mu + k_{ve}$$

The above mass balance relations were derived assuming that (i) the size of the cytosolic pool of CAA is relatively small

compared with the size of the vacuolar and protein CAA pool, that is, $S_1 \ll S_2 + S_3$, and (ii) CAA are not significantly degraded or exported (Fig S8A and B). The consensus weights of the protein and vacuolar pools $S_2$ and $S_3$ were determined as described in "Determination of transport rates and sizes of amino acid pools" using labeling of CAA in soluble and protein pools at each time point $f_{prot}(t)$ and $f_{sol}(t)$ (Equations (4) and (5)) and were averaged across the time points for each culture. For calculation purposes the weight of the cytosolic pool was set to an arbitrarily small value $S_1 = 0.01$.

The free model parameters included $\kappa_{pd}$, $\mu$, and $\kappa_{ve}$ (which affect the values in the **M**-matrix) and were quantified by fitting the model with experimental labeling values for protein and soluble pools using a nonlinear least-squares solver:

$$[k^\star_{pd}, \mu^\star, k^\star_{ve}] = \min_{k_{pd},\mu,k_{ve}} {}_i \left( \left( f_{prot}(t_i) - [0,1,0]e^{\mathbf{M}t_i}\mathbf{1}_n \right)^2 + \left( f_{sol}(t_i) - [0,0,1]e^{\mathbf{M}t_i}\mathbf{1}_n \right)^2 \right)$$

The fitted parameters were constrained to reflect the fact that: (i) vacuolar export rates can have a wide range; (ii) bulk protein degradation rates are low compared with the growth rates of yeast cell cultures under optimal conditions (Wiechecki et al, 2017 Preprint); and (iii) a reasonable growth rate limit for budding yeast cultures is ~0.45 h⁻¹, which corresponds to ~1.5 h doubling time:

$$0.01\,h^{-1} \le k_{ve} \le 10\,h^{-1}$$
$$0.01\,h^{-1} \le k_{pd} \le 0.05\,h^{-1}$$
$$0.3\,h^{-1} \le \mu \le 0.45\,h^{-1}$$

Notably, $\mu$ was kept as a free optimization parameter as it could not be determined nearly as accurate the labeling values using turbidity measurements. Nevertheless, to test the validity of our results, we also have set $\mu$ to the experimental growth rates derived from turbidity measurements observing the same vacuolar export patterns. The fitting and extraction of the model parameters including the vacuolar export rate and visualization of the fits were performed using a custom Python script (see the Data Availability section). Vacuolar export rates we converted to units of the protein pool size as:

$$K_{ve} = \frac{k_{ve}S_3}{S_2}$$

## Data Availability

The source data and scripts used in this study to analyze lysine and arginine vacuolar transport using dynamic labeling assays have been deposited to the Zenodo repository (Onischenko & Noor, 2026).

## Supplementary Information

# Acknowledgements

We thank Christos Gournas for critical reading of the manuscript, regular exchanges, and mentorship and members of the Molecular Physiology of the Cell laboratory for fruitful discussions. E Zaremba is a fellow of the Fonds pour la formation à la Recherche dans l'Industrie et dans l'Agriculture (FRIA), the Fonds David et Alice Van Buuren, the Fondation Jaumotte-Demoulin, and the International Brachet Foundation. This work was supported by the Fonds Stimulus from Meurice R&D and by a research grant from the Research Council of Norway (NFR 315615) to E Onischenko and E Noor.

### Author Contributions

E Zaremba: conceptualization, formal analysis, validation, investigation, visualization, and writing—original draft, review, and editing.
F Vierendeels: validation and investigation.
R Dutoit: formal analysis, investigation, visualization, and writing—original draft, review, and editing.
E Bodo: formal analysis, validation, investigation, and writing—review and editing.
E Bifulco: formal analysis, and writing—original draft, review, and editing.
C Tricot: Resources, funding acquisition, validation, and writing—review and editing.
E Noor: formal analysis, funding acquisition, validation, investigation, and writing—original draft, review, and editing.
B André: conceptualization, resources, funding acquisition, and writing—review and editing.
E Onischenko: conceptualization, formal analysis, funding acquisition, validation, investigation, visualization, and writing—original draft, review, and editing.
M Cools: conceptualization, formal analysis, supervision, funding acquisition, investigation, visualization, project administration, and writing—original draft, review, and editing.

### Conflict of Interest Statement

The authors declare that they have no conflict of interest.

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
