## [Reviewer comments · Life Science Alliance]

Vsb1, Ypq1 and Ypq2 control dynamic cationic amino acid storage in the yeast vacuole

Evi Zaremba, Fabienne Vierendeels, Raphaël Dutoit, Elisabeth Bodo, Ersilia Bifulco, Catherine Tricot, Elad Noor, Bruno André, Evgeny Onischenko, and Melody Cools

DOI: <https://doi.org/10.26508/lsa.202503520>

Corresponding author(s): Melody Cools, Labiris

Review Timeline:

Submission Date:	2025-10-07
Editorial Decision:	2025-11-20
Revision Received:	2026-03-17
Editorial Decision:	2026-04-13
Revision Received:	2026-04-17
Accepted:	2026-04-27

Scientific Editor: Sarita Hebbar

Transaction Report:

November 20, 2025

Re: Life Science Alliance manuscript #LSA-2025-03520-T

Dr. Melody Cools
Labiris
1 avenue Emile Gryzon
Brussels 1070
Belgium

Dear Dr. Cools,

Thank you for submitting your manuscript entitled "Vsb1, Ypq1 and Ypq2 control dynamic cationic amino acid storage in the yeast vacuole" to Life Science Alliance. Your manuscript was evaluated by three expert reviewers whose comments are appended below. All three reviewers agree that this work is of potential interest to the field. You will also note that the reviewers had several concerns that preclude publication at this stage.

Most of the reviewers' concerns are connected to an overstatement of the results, the need for better representation of figures, the lack of contextualisation of the findings to state of the knowledge in the field, and missing methodological details. We agree with the reviewers that all such concerns must be addressed. In this context we agree with Reviewer 1 that statements not supported by accompanying data must be either toned down or you must provide evidence for the same.

Although optional, we strongly encourage you to provide the following information:

1. data on vsb1 Δ cells with a clean genetic background and in comparison with WT as recommended by Reviewer 1.
2. supplement the data on localisation of Lyp1-GFP with information on functionality as suggested by Reviewer 3, point on figure 2.
3. in agreement with valid point of Reviewer 2 on the presented conservation analyses, include the sequence alignment of these 3 residues across homologs.

In line with their overall assessment, we invite you to submit a revised manuscript addressing the reviewers' comments. When submitting the revision, please include a letter addressing the reviewers' comments point by point. While a rebuttal must respond to all points in some form, additional experiments to resolve these points, other than indicated above, will not be required

I would be happy to discuss the revision in more detail via email or phone/videoconferencing. Please let me know which option you prefer, if any.

While you are revising your manuscript, please also attend to the below editorial points to help expedite the publication of your manuscript. Please direct any editorial questions to the journal office. When submitting the revision, please include a letter addressing the reviewers' comments point by point.

Thank you for this interesting contribution to Life Science Alliance. We hope that the comments below will prove constructive as your work progresses, and we are looking forward to receiving your revised manuscript.

Sincerely,

Sarita Hebbar, PhD
Scientific Editor
Life Science Alliance
<http://www.lsjournal.org>

B. MANUSCRIPT ORGANIZATION AND FORMATTING:

Reviewer #1 (Comments to the Authors (Required)):

In this study, Zaremba et al investigate how yeast cells might scavenge and assimilate amino acids, using the vacuole as a store/source. The current study explores the transport, storage and mobilization of cationic amino acids (CAA), like lysine and arginine at the vacuole. Specifically, the authors identify the transporters for lysine at the vacuolar membrane and understand how different transporters work together to maintain CAA homeostasis. To address this, the authors use a combination of modeling, biochemical transport assays and mutagenesis experiments and assign the Vsb1 and Ypq1 transporters respectively as the vacuolar importer and exporter of lysine.

The identification of the Vsb1 and Ypq1 transporters in this process is clear, and exciting! However, the study diverges from a central point, and in many places, there is a substantial overstatement of novelty. This study is more correctly a quantitative dissection of known transporters (as opposed to the 'discovery' of a new type of transporter etc), and their potential role in balancing aa pools inside the vacuole/vs in the cytoplasm. There are a few overstatements, and some key data that are missing (based on their inferences).

Specific comments:

1. Lines 57-77: The extensive detail on arginine transport is not in any way relevant to the current story. The information here can be reduced to key points.
2. Lines 78-84: Similarly, how is this extensive description of lysine biosynthesis pathway important or relevant for the current story? This is distracting away from main point.
3. Lines 95-110: there are published reports which have identified vacuolar lysine transporters. These have done so by providing genetic evidence, and using transport assays and compartment analysis much like the current study. Therefore, these studies need to be properly contextualized with reference to this study.
4. Line 113: In continuation to the previous comment, how much is this study different? The novelty in this study (from my reading) actually lies in understanding relative contributions of these transporters for import/v/export (and not in their identification). Please elaborate accordingly.
5. Line 167: This data can be moved to supplementary.
6. Lines 176-177: Fig. 2A shows the total soluble lysine pool in wt and vsb1 . No data for reduced uptake is shown. A reduced pool cannot be equated to reduced uptake, without supporting data (since a reduction in the total pool can be a result of multiple distinct possibilities - production, consumption, uptake etc).
7. Lines 180-184: Can the cytosolic and vacuolar lysine pools of wt and vsb1 be measured and quantified? It is relatively straightforward to isolate intact vacuoles from yeast (although there are challenges given the time of processing), and this should clarify a quantitative argument. Also, please explain the rationale behind using *rsp5* background in these experiments.
8. Fig 2B compares the total soluble pool, and does not measure the uptake rates. There is no data provided on uptake rates/kinetics of any kind.

9. Lines 190-194: Why is this experiment also done in a *rsp5* background? What happens in *wt* vs *vsb1* scenario (clean genetic background)?
10. Lines 201-203: The rationale of looking at reduced uptake can probably come in this section, since Fig. 3A suggests reduced uptake in the *vsb1* cells for the first time.
11. Line 203: The data does not support the statement "the initial uptake rates remain similar....", since there is a clear decrease in uptake in the *vsb1* cells as shown in fig 3A. How was this result inferred?
12. There is some disjointed presentation of the data - the entire section on uptake is more related to Fig 3 (and not really related to Fig 2 and associated text/data). A reorganization of this data accordingly will make the characterization clearer.
13. Data in Fig 6 strongly suggests that these transporters may also be involved in dynamic CAA exchange even in nutrient replete conditions. However, this entire section (which is probably the highlight of this study) is presented in a very dense manner and difficult to fully understand. This is also a bit confounding because of the three different strains used. It can probably be broken down into two distinct figures, and should be clarified better. For instance, more details on how to interpret the data in panels C-G.

Reviewer #2 (Comments to the Authors (Required)):

In this manuscript, Zaremba et al. present a well-structured and compelling study identifying *Vsb1* as a vacuolar lysine importer, *Ypq1* as a vacuolar lysine exporter, and *Ypq2* as a vacuolar arginine exporter. These conclusions strongly supported by the data, and the experimental evidence is sufficient to support the overall conclusion. This work provides important mechanistic insights into vacuolar amino acid transport. Nevertheless, several figures would benefit from improved formatting, and additional methodological details would clarify the study and will help future investigations that build off of these findings.

Major Comments

1. Figure 6 presentation

- 6C: The growth dilution curve is not clearly presented. Because the growth data and \log_2 light fraction curves use different units, they should not be plotted on the same graph. I suggest either moving the growth curves to the supplementary figures or displaying them separately, for example, on a secondary y-axis.
- 6D, E, F, G, I: The tables should be reformatted so that the text does not overlap with the table borders. Additionally, the current presentation of p-values is somewhat confusing. A clearer layout-such as a more conventional table design (1D array)- would help make this data easier to interpret.

2. Additional methodological details are needed in the Materials and Methods section.

- Lines 670-671 (and elsewhere): The description "Data are reported as mean values, with error bars representing standard deviations" should be clarified. It would be helpful to specify what the means and standard deviations represent-e.g., biological, technical, or analytical replicates-for transparency and readability.
- Mass spectrometry methods:
 - i. Please clarify if the LC gradient is correctly described. It does not appear to be a standard design, for example, there seem to be little to no equilibration of the column (equilibration needs to be held for several column volumes at a fixed solvent concentration). Moreover, no raw chromatographic data are shown for either standards or biological samples. Including representative chromatograms would help demonstrate how the LC-MS quantification was performed.
 - ii. The isolation window was set to 17 m/z, which exceeds the mass difference between the light and heavy isotope-labeled lysines and risks co-isolation of ions. Additional context on why this window was chosen would be valuable.
 - iii. Lines 866-869: More details on calibrator preparation are needed. Please specify the exact concentrations of the calibrators and whether this was an isotope-dilution calibration curve. If so, indicate whether heavy or light Arg/Lys was used as the internal standard. Including the calibration curve in the Results section would also help illustrate the quantification approach.

3. Conservation analysis

- Figure Supplement 2 nicely shows the structural positioning of T227, Q275, and Q278 relative to other transporters. To further support the interpretation, it would be helpful to include a sequence alignment to demonstrate conservation of these residues across homologs.

Minor Comments

1. The notation for ratios is inconsistent and could be clearer. Consider replacing the period (".") with the multiplication symbol ("•") or using A/B instead of $A \cdot B^{-1}$ to improve readability.
2. Figure 1: Do the mutants display any growth defects? It would be helpful to clarify whether *Ypq1* deletion or *Vsb1* overexpression has any toxic effects.
3. Figure 2A: In addition to clarifying how standard deviations were calculated, it would be helpful to address why the *vsb1* Δ condition does not have error bars. If this was unintentional, the figure should be revised.

Overall, this is a strong and well-executed study. With clearer figure formatting and additional methodological detail, the study will be more accessible to the broader research community.

Reviewer #3 (Comments to the Authors (Required)):

The manuscript by Zaremba et al. describes the regulatory mechanisms used by yeast cells to control the storage and availability of the cationic amino acids lysine and arginine in yeast cells. The authors describe the vacuolar protein Vsb1 as the main source for lysine import into the vacuole in a proton gradient dependent process. They furthermore identify Ypq1 as the main vacuolar protein that exports lysine from the vacuole in a proton gradient independent process, while arginine is mainly transported by Ypq2. The authors use a variety of methods ranging from amino acid measurements, structural predictions and functional analysis as well as metabolic flux labeling to determine the complex interplay of these processes. Overall, this is a well-designed study that carefully analyzes the complex interplay between CAA uptake over the plasma membrane, storage in the vacuole and replenishing of emptied CAA stores. All the experiments are carefully designed and controlled and proper statistics are available for all experiments. The paper is well written but some parts could be streamlined and condensed. Both the introduction and the discussion are very long and the information could be delivered in a shorter form. The manuscript is interesting for the readership of Life Science Alliance and is of interest for the yeast transport field community and researchers working on amino acid metabolism. I think the manuscript should be published with a few minor points that can be raised:

Figure 1: What is the authors' explanation for the reduced soluble pool of lysine in the ypq1-2-3 mutant compared to the ypq1 mutant alone? Is this just an effect of the different sample sizes or is there a biological explanation for this point?

Figure 2: Many plasma membrane transporters are very difficult to tag with a GFP. While the localization of Lyp1-GFP looks perfectly fine, it would be good to show that this construct is also functional. This should be easily testable by combining the Lyp1-GFP construct with a lys2 Δ mutant to test viability.

Figure 6: The data in figure 6 are very difficult to interpret. The largely different scales in fig 6D are somewhat confusing. While a change in -0.1 might be significant, is it meaningful compared to the large changes in soluble CAAs? It could be more informative to plot all plots with the same scale, as this would more realistically reflect the minor changes in the protein and the total pool.

Discussion: Several reports have shown that both lysine and arginine can be used in prototrophic yeast cells to label the proteins for MS analysis (doi: 10.1021/acs.analchem.8b02557.; doi: 10.1074/mcp.M112.025742.). Maybe the authors could incorporate a short statement on how the vacuolar storage capacity could affect such methods.

Sarita Hebbar, PhD
Scientific Editor
Life Science Alliance
<http://www.lsjournal.org>

Dear Dr. Hebbar,

Please find below a point-by-point list of the revisions made to address the issues raised by the reviewers for our manuscript entitled "Vsb1, Ypq1 and Ypq2 control dynamic cationic amino acid storage in the yeast vacuole". Original comments are in black, replies in blue.

We would like to thank you for your helpful summary and positive attitude regarding our work. We believe that we have adequately addressed all the points raised and would like to thank the reviewers for their interest in our work and their thoughtful and spot-on comments that significantly improved the manuscript.

In particular, we have focused on the following concerns:

1. We provide data related to ^{14}C -lysine accumulation in the vacuolar fractions of the *rsp5* and the *rsp5vsb1Δ* strains as these were previously carried out with strains harbouring an additional *car1Δ* mutation. This had historically been needed for work focused on arginine transport. However, we explain in our response to reviewer 1 why these experiments need to be performed in an *rsp5* background (see below) and present this data in a different order with additional experiments to make this point more evident for future readers.
2. We have added data showing that the Lyp1-GFP fusion is functional.
3. We have added a sequence alignment of the transmembrane domains of Vsb1 and of members of the SLC26A family.

In addition, the section concerning dynamic vacuolar cationic amino acid exchange has been rewritten in a more didactic manner, expanding it to 3 smaller chapters that explain our approach in more details.

We hope that the revised manuscript is now acceptable for publication in *Life Science Alliance*.

Sincerely,

Dr. Melody Cools

Reviewer #1 (Comments to the Authors (Required)):

In this study, Zaremba et al investigate how yeast cells might scavenge and assimilate amino acids, using the vacuole as a store/source. The current study explores the transport, storage and mobilization of cationic amino acids (CAA), like lysine and arginine at the vacuole. Specifically, the authors identify the transporters for lysine at the vacuolar membrane and understand how different transporters work together to maintain CAA homeostasis. To address this, the authors use a combination of modeling, biochemical transport assays and mutagenesis experiments and assign the Vsb1 and Ypq1 transporters respectively as the vacuolar importer and exporter of lysine.

The identification of the Vsb1 and Ypq1 transporters in this process is clear, and exciting! However, the study diverges from a central point, and in many places, there is a substantial overstatement of novelty. This study is more correctly a quantitative dissection of known transporters (as opposed to the 'discovery' of a new type of transporter etc), and their potential role in balancing aa pools inside the vacuole/vs in the cytoplasm. There are a few overstatements, and some key data that are missing (based on their inferences).
Specific comments:

1. Lines 57-77: The extensive detail on arginine transport is not in any way relevant to the current story. The information here can be reduced to key points.

We thank the reviewer for their comment. We agree that too much detail on vacuolar arginine transport was provided so we have trimmed this paragraph to only essential information. Extensive information regarding the SLC26A/SulP family has been deleted as well since it appears later in detail in the result section. The text has been modified to focus less on arginine transport.

2. Lines 78-84: Similarly, how is this extensive description of lysine biosynthesis pathway important or relevant for the current story? This is distracting away from main point.

We have deleted most information pertaining to the lysine biosynthesis pathway as we agree with reviewer 1 that it distracts from the main points.

3. Lines 95-110: there are published reports which have identified vacuolar lysine transporters. These have done so by providing genetic evidence, and using transport assays and compartment analysis much like the current study. Therefore, these studies need to be properly contextualized with reference to this study.

We agree with the reviewer that previously published data needs to be more accurately presented in the text in order to reflect more equitably what the present study contributes to the field. These lines have thus been rewritten to reflect more precisely what each study has reported about vacuolar lysine transporters.

4. Line 113: In continuation to the previous comment, how much is this study different? The novelty in this study (from my reading) actually lies in understanding relative contributions of these transporters for import/v/export (and not in their identification). Please elaborate accordingly.

We thank the reviewer for correctly assessing the novelty of our study and pointing out that it should be contextualized properly with regards to other similar work. We have changed the text to reflect that we set out to identify the primary vacuolar lysine importer and exporter among several known vacuolar lysine transporters in yeast and to investigate the contribution of vacuolar CAA transporters to CAA homeostasis. We have highlighted our finding that, contrary to previously reported, Ypq1 seems to function as the primary lysine exporter rather than a lysine importer. Finally, we have also emphasized that this is the first study to evaluate the kinetics of vacuolar exchange of lysine and arginine directly in live cells.

5. Line 167: This data can be moved to supplementary.

We appreciate the view of the reviewer, and we agree that results pertaining to the epistasis relationship between Vsb1 and Ypq1 is not the focus of that section. However, we believe it would be easier for future readers to keep the total amino acid pool measurements in one figure rather than separate them in 2 distinct figures, with one being isolated in a supplementary figure.

6. Lines 176-177: Fig. 2A shows the total soluble lysine pool in wt and vsb1 Δ . No data for

reduced uptake is shown. A reduced pool cannot be equated to reduced uptake, without supporting data (since a reduction in the total pool can be a result of multiple distinct possibilities - production, consumption, uptake etc).

The reviewer is correct that Fig. 2A does not show uptake measurements but how the total lysine pools of the w-t and the *vsb1Δ* strains are affected by the supplementation of lysine in the medium. We agree that an increase in total lysine content could also stem from increased lysine production or decreased consumption, although the former would be unlikely as lysine synthesis is inhibited in the presence of external lysine. To avoid this misrepresentation, we have deleted the word “accumulation” in this section and now only mention specifically total lysine pools.

7. Lines 180-184: Can the cytosolic and vacuolar lysine pools of wt and *vsb1Δ* be measured and quantified? It is relatively straightforward to isolate intact vacuoles from yeast (although there are challenges given the time of processing), and this should clarify a quantitative argument. Also, please explain the rationale behind using *rsp5Δ* background in these experiments.

We thank the reviewer for this excellent point. A figure has been added (Fig. 1 B) that shows compartmental distribution of lysine in w-t, *vsb1Δ*, *ypq1Δ*, *VSB1* OE and *YPQ1* OE strains. This data confirms that over 85% of the total lysine pool of the cell remains vacuolar in those strains with a slight reduction of vacuolar pools if *VSB1* is deleted or *YPQ1* is overexpressed. This is consistent with a role of Vsb1 and Ypq1 in vacuolar lysine compartmentalization.

The reasoning for using the *rsp5* mutant is explained below in comment 9.

8. Fig 2B compares the total soluble pool, and does not measure the uptake rates. There is no data provided on uptake rates/kinetics of any kind.

The reviewer is right that we do not provide uptake rates or kinetics regarding Vsb1 activity. Fig 2B (now Fig 2E) shows measurements of accumulated ¹⁴C-lysine and its derivatives after 1 hour of supplementation with ¹⁴C-lysine for the *rsp5* and *rsp5 vsb1Δ* strains and *rsp5* strain treated with bafilomycin A. It is a control experiment to check that, with adjusted concentrations of radiolabeled lysine, we can achieve similar accumulation levels of ¹⁴C-lysine across strains and conditions after 1 hour (granted there are some statistically significant differences).

9. Lines 190-194: Why is this experiment also done in a *rsp5Δ* background? What happens in wt vs *vsb1Δ* scenario (clean genetic background)?

We agree that these experiments need to be done in a clean genetic background. As they were carried out with an *rsp5car1Δ* strain and *rsp5car1Δvsb1Δ* strain that were previously used for arginine transport studies, we have redone the experiments in *rsp5* and *rsp5vsb1Δ* mutants. Unfortunately, we are unable to do this permeabilization experiment in a strain lacking the *rsp5* mutation as, in such a strain, Lyp1 would be endocytosed, and accumulation of the radiolabeled lysine would be low, especially for the *vsb1Δ* strain. This would lead to a poor signal-to-noise ratio for vacuolar and cytosolic pools which are diluted during the course of permeabilization and extraction. In addition, it would make achieving comparable accumulation levels of ¹⁴C-lysine across strains more difficult. To illustrate how the *rsp5* mutation affects ¹⁴C-lysine accumulation, we have added data showing that mutating *RSP5* in the *vsb1Δ* strain enhances ¹⁴C-lysine uptake by 2-fold (Fig 2B).

10. Lines 201-203: The rationale of looking at reduced uptake can probably come in this section, since Fig. 3A suggests reduced uptake in the *vsb1Δ* cells for the first time.

We agree with the reviewer and have shuffled experiments so that uptake experiments come earlier.

11. Line 203: The data does not support the statement "the initial uptake rates remain similar...", since there is a clear decrease in uptake in the *vsb1Δ* cells as shown in fig 3A. How was this result inferred?

Indeed, there is a small non-statistically significant reduction in initial uptake rate. We have done a new set of experiments to include the *rsp5* and *rsp5vsb1Δ* mutants. With this new set, the difference between the w-t and the *vsb1Δ* mutant is less apparent. We have modified the text accordingly and have added more details in the materials and methods section on how initial uptake rates are calculated.

12. There is some disjointed presentation of the data - the entire section on uptake is more related to Fig 3 (and not really related to Fig 2 and associated text/data). A reorganization of this data accordingly will make the characterization clearer.

We have reorganized the data and hope the reviewer will find it more clearly presented.

13. Data in Fig 6 strongly suggests that these transporters may also be involved in dynamic CAA exchange even in nutrient replete conditions. However, this entire section (which is probably the highlight of this study) is presented in a very dense manner and difficult to fully understand. This is also a bit confounding because of the three different strains used. It can probably be broken down into two distinct figures, and should be clarified better. For instance, more details on how to interpret the data in panels C-G.

We thank this reviewer for the valuable feedback regarding the description of dynamic labeling assays. Following this advice, we re-wrote the whole section in a more didactic manner explaining the motivation behind the experimental design and better describing the interpretations. Specifically, we have split the whole chapter into three smaller sections. We begin with conceptual description of steady-state labeling dynamics, its utility to detect vacuolar export and implementation of this framework for detecting CAA export in wild type cells. In the second section we describe a more complex case of determining roles *YPQ* genes in lysine and arginine transport using comparative analysis of dynamic labeling in subcellular fractions in three *ypqΔ* mutants and w-t cells. We have now split the figure 6 into two smaller ones (now Fig 5 and Fig 6) and use Figure 5 (panels C-E), and extended text to better explain how comparative analysis of *ypq* mutants were used to dissect the role of these genes in CAA transport. Last, in the third section and Figure 6 we focus entirely on the quantitative analysis of the vacuolar transport capitalizing on our accurate dynamic labeling measurements. We hope that in this manner we convey more clearly the reason of using three *ypqΔ* mutants.

Reviewer #2 (Comments to the Authors (Required)):

In this manuscript, Zaremba et al. present a well-structured and compelling study identifying *Vsb1* as a vacuolar lysine importer, *Ypq1* as a vacuolar lysine exporter, and *Ypq2* as a vacuolar arginine exporter. These conclusions strongly supported by the data, and the experimental evidence is sufficient to support the overall conclusion. This work provides important mechanistic insights into vacuolar amino acid transport. Nevertheless, several figures would benefit from improved formatting, and additional methodological details would clarify the study and will help future investigations that build off of these findings.

Major Comments

1. Figure 6 presentation

- 6C: The growth dilution curve is not clearly presented. Because the growth data and \log_2 light fraction curves use different units, they should not be plotted on the same graph. I suggest either moving the growth curves to the supplementary figures or displaying them separately, for example, on a secondary y-axis.

We thank the reviewer for this comment. Although we kindly disagree that growth dilution and the light fraction have different units (they are both fractions, i.e. unitless values) we agree that conveying exponential growth based on growth dilution is confusing. We therefore follow this reviewer advice and now show log-transformed growth dynamics in a separate supplemental figure S7B. We also clarify in the text that growth dilution is evaluated **based** on the culture growth dynamics (i.e. as the inverse) and in the main figure (now Fig. 5C) show all curves as raw fractions to present these results in a way more coherent with the explanation of the experimental design as illustrated in the Fig. 5B.

- 6D, E, F, G, I: The tables should be reformatted so that the text does not overlap with the table borders. Additionally, the current presentation of p-values is somewhat confusing. A clearer layout-such as a more conventional table design (1D array)-would help make this data easier to interpret.

Following this suggestion we have aligned the tables and included additional 1D lists of p-values for all the analyses in a separate spreadsheet (Supplemental data 1).

2. Additional methodological details are needed in the Materials and Methods section.

- Lines 670-671 (and elsewhere): The description "Data are reported as mean values, with error bars representing standard deviations" should be clarified. It would be helpful to specify what the means and standard deviations represent-e.g., biological, technical, or analytical replicates-for transparency and readability.

We agree with the reviewer that what mean values and standard deviations represent should be specified and have added corresponding descriptions to figure legends. We specify that our statistical analyses and error estimations are based on biological replicates.

- Mass spectrometry methods:

i. Please clarify if the LC gradient is correctly described. It does not appear to be a standard design, for example, there seem to be little to no equilibration of the column (equilibration needs to be held for several column volumes at a fixed solvent concentration). Moreover, no raw chromatographic data are shown for either standards or biological samples. Including representative chromatograms would help demonstrate how the LC-MS quantification was performed.

We have added details of the column equilibration to the "Mass spectrometry analysis" section in materials and methods. "For equilibration, the B concentration was turned back to 85 % over 4 min. This equilibration time was decided to ensure stable baseline conditions and reproducible retention times. Additionally, the injection needle was washed for 2 minutes with 50% methanol while 85% B was continuously flushed onto the column (approximate 5 column volumes in total)."

We have also included representative raw chromatograms of amino acids monitored with our LC-MS setup (Fig S9) as well as exemplary chromatograms corresponding to the analysis of CAA in the dynamic labelling assays (Fig S6).

ii. The isolation window was set to 17 m/z, which exceeds the mass difference between the light and heavy isotope-labeled lysines and risks co-isolation of ions. Additional context on why this window was chosen would be valuable.

We added a relevant justification to “Mass spectrometry analysis” section of materials and methods. Specifically, we explain that the wide 17 m/z window was intentionally used for the Orbitrap targeted SIM (tSIM) scans to simultaneously analyze all relevant compounds with similar mass in the same mass scan. Since tSIM is an MS1 level acquisition method that does not involve fragmentation, the high resolving power 70,000 of the Orbitrap is sufficient to cleanly resolve all amino acids targeting them at their precise m/z values. The benefit of this analysis is that the isotopologues are analyzed at identical conditions making ratiometric quantification highly accurate in a wide range of concentrations. In Fig S9 we provide exemplary chromatograms for simultaneous monitoring of 12 amino acids (including the ones used in this study) both in a standard mix and in a w-t cell extract.

Supplemental data 2 (Supplemental data 2, inclusion list) contains the complete inclusion list used to simultaneously monitor amino acids and their isotopologues.

iii. Lines 866-869: More details on calibrator preparation are needed. Please specify the exact concentrations of the calibrators and whether this was an isotope-dilution calibration curve. If so, indicate whether heavy or light Arg/Lys was used as the internal standard. Including the calibration curve in the Results section would also help illustrate the quantification approach.

We apologize for the lack of clarity about our analytical setup. To clarify how readouts of our dynamic labeling assays – fraction of the light CAA, f , - were quantified from LC-MS data and how equimolar standards were used, we collected all explanations regarding this at one place at in the “Metabolic labeling assays” section of materials and methods. Specifically, we emphasize that light CAA fraction in our samples is always defined by the **ratio** of light and heavy isotopologue intensities (in other words, our assays are internally referenced where light and heavy variants play role of a standard for each other). To explicitly show this we rearranged the original quantification formula as:

$$f = \frac{w_L I_L}{w_L I_L + w_H I_H} = \frac{1}{1 + (w_H I_H)/(w_L I_L)}$$

where we can see our experimental readouts are dependent on heavy/light intensity ratios. We further provide exact concentrations of the equimolar mixes of heavy/light isotopologues used to quantify their relative signal response factors W_H and W_L (i.e., the intensity adjustment coefficients for light and heavy amino acid variants), and additionally provide their quantification formulae:

$$w_H = \frac{1}{1 + I_L^*/I_H^*} ; w_L = \frac{1}{1 + I_H^*/I_L^*}$$

To demonstrate robustness of light/heavy ratios quantification over a wide range of concentrations (covering our assays), we show representative chromatograms of heavy/light standard mixture (Fig S6A) and provide results of heavy/light ratio quantifications in serial dilution injections (see Supplemental data 2, calibration standards). Of note, the high stability of the light/heavy ratios is not a surprise since both isotopologues are analyzed simultaneously.

3. Conservation analysis

- Figure Supplement 2 nicely shows the structural positioning of T227, Q275, and Q278 relative to other transporters. To further support the interpretation, it would be helpful to include a sequence alignment to demonstrate conservation of these residues across homologs.

We thank the reviewer for this insightful suggestion. We have added a structural-based alignment of the TM domain of Vsb1 with different SLC26A transporters (Figure S3).

Minor Comments

1. The notation for ratios is inconsistent and could be clearer. Consider replacing the period (":") with the multiplication symbol ("•") or using A/B instead of $A \bullet B^{-1}$ to improve readability.

We agree that notations need to be more consistent and have replaced the period with A/B for improved readability.

2. Figure 1: Do the mutants display any growth defects? It would be helpful to clarify whether Ypq1 deletion or Vsb1 overexpression has any toxic effects.

We thank the reviewer for raising this point. Deletion and overexpression strains did not show any growth defect in minimal ammonium/glucose medium. This information was added to the manuscript (Table S1).

3. Figure 2A: In addition to clarifying how standard deviations were calculated, it would be helpful to address why the *vsb1Δ* condition does not have error bars. If this was unintentional, the figure should be revised.

We agree with the reviewer that not having error bars for the *vsb1Δ* mutant would be confusing for readers. There are error bars for the *vsb1Δ* mutant but they are not visible at this scale, especially compared to the w-t data. A note clarifying this has been added to the legend.

Overall, this is a strong and well-executed study. With clearer figure formatting and additional methodological detail, the study will be more accessible to the broader research community.

Reviewer #3 (Comments to the Authors (Required)):

The manuscript by Zaremba et al. describes the regulatory mechanisms used by yeast cells to control the storage and availability of the cationic amino acids lysine and arginine in yeast cells. The authors describe the vacuolar protein Vsb1 as the main source for lysine import into the vacuole in a proton gradient dependent process. They furthermore identify Ypq1 as the main vacuolar protein that exports lysine from the vacuole in a proton gradient independent process, while arginine is mainly transported by Ypq2. The authors use a variety of methods ranging from amino acid measurements, structural predictions and functional analysis as well as metabolic flux labeling to determine the complex interplay of these processes. Overall, this is a well-designed study that carefully analyzes the complex interplay between CAA uptake over the

plasma membrane, storage in the vacuole and replenishing of emptied CAA stores. All the experiments are carefully designed and controlled and proper statistics are available for all experiments. The paper is well written but some parts could be streamlined and condensed. Both the introduction and the discussion are very long and the information could be delivered in a shorter form. The manuscript is interesting for the readership of Life Science Alliance and is of interest for the yeast transport field community and researchers working on amino acid metabolism. I think the manuscript should be published with a few minor points that can be raised:

We thank the reviewer for their positive assessment of our work. As reviewer 1 has also commented on the introduction being too long, superfluous information has been deleted.

Figure 1: What is the authors' explanation for the reduced soluble pool of lysine in the *ypq1-2-3* mutant compared to the *ypq1* mutant alone? Is this just an effect of the different sample sizes or is there a biological explanation for this point?

We thank the reviewer for their observation. Indeed, there is a difference in total lysine pool size between the *ypq1Δ* mutant and the triple *ypq1-2-3Δ* mutant. Sample sizes could account for this difference as fewer biological replicates were analyzed for the triple *ypq1-2-3Δ* mutant compared to the single mutant. Additionally, secondary regulatory effects apparent only in the triple mutant could possibly lead to a difference in pool size. This includes but is not limited to regulation of lysine biosynthesis and regulation of transporter activity. Another possible explanation is that Ypq2 and Ypq3 could transport lysine to some degree. Based on the data presented in Fig 6, this is not unlikely for Ypq2.

Figure 2: Many plasma membrane transporters are very difficult to tag with a GFP. While the localization of Lyp1-GFP looks perfectly fine, it would be good to show that this construct is also functional. This should be easily testable by combining the Lyp1-GFP construct with a *lys2Δ* mutant to test viability.

We agree with the reviewer that Lyp1-GFP activity needs to be assessed. We have measured initial uptake rates of ¹⁴C-lysine in a strain unable to transport lysine (*gap1Δcan1Δlyp1Δ*) and the same strain expressing Lyp1 w-t or the Lyp1-GFP fusion from a plasmid. The triple deletion strain is unable to transport lysine while the strains expressing the tagged and untagged version of Lyp1 both transport lysine, albeit with different initial uptake rates. The fact that Lyp1-GFP is functional was added to the manuscript.

Figure 6: The data in figure 6 are very difficult to interpret. The largely different scales in fig 6D are somewhat confusing. It could be more informative to plot all plots with the same scale, as this would more realistically reflect the minor changes in the protein and the total pool.

We entirely agree with the reviewer that the description of dynamic labeling assays suffered from the lack of clarity. In the revised version we provide a more detailed description of the experiments and their interpretations (including what is now Fig 5E). We hope that the comparative analysis of labeling dynamics in subcellular fractions is now more clearly explained providing evidence of *YPQ1* and *YPQ2* selectivity towards lysine and arginine transport. We nevertheless kindly disagree with the suggestion to rescale the panels as the mirror effects of mutations in the renewal of in the protein and soluble fractions would not be easy to convey in this case.

While a change in -0.1 might be significant, is it meaningful compared to the large changes in soluble CAAs?

In our qualitative analysis of CAA renewal dynamics in *ypq* mutants (Fig 5E and Fig S7C) we are primarily concerned with the sign of the difference within each fraction (more or less) rather than comparing the magnitudes. Magnitudes are compared only in one case to support our conclusion that changes in the renewal dynamics in the vacuolar and protein pools compensate for each other. We included the following sentence to clarify this interpretation: “Most strikingly, we observed that slower renewal of lysine and arginine in the vacuolar pools was always accompanied by faster renewal in the protein pool (Fig 5E). The balance of these effects was supported by smaller mutation-related effects in the total lysates that represent both pools combined as compared to the individual pools (Fig 5E).”

Discussion: Several reports have shown that both lysine and arginine can be used in prototrophic yeast cells to label the proteins for MS analysis (doi: 10.1021/acs.analchem.8b02557.; doi: 10.1074/mcp.M112.025742.). Maybe the authors could incorporate a short statement on how the vacuolar storage capacity could affect such methods.

We thank the reviewer for this excellent point. Judging by the significant sizes of the vacuolar CAA pools and their dynamic exchange with the cytosolic pool mediated by *YPQ1* /2, as evidenced by our analysis, these dynamic stores are expected to greatly affect the dynamics of protein labeling in dynamic SILAC experiments used to monitor stability and degradation kinetics of cellular proteins. Therefore, results of proteomics studies using SILAC must factor in these effects for correct interpretation. We have alluded on this point in the last paragraph of revised discussion.

April 13, 2026

RE: Life Science Alliance Manuscript #LSA-2025-03520-TR

Dr. Melody Cools
Labiris
1 avenue Emile Gryzon
Brussels 1070
Belgium

Dear Dr. Cools,

Thank you for submitting your revised manuscript entitled "Vsb1, Ypq1 and Ypq2 control dynamic cationic amino acid storage in the yeast vacuole".

Your revised manuscript was evaluated by two of the three original reviewers. They have both commented that the revised manuscript addressed their previous concerns and is significantly improved.

We would be happy to publish your paper in Life Science Alliance pending resolution of final revisions necessary to meet our formatting guidelines. We request you to submit a revised manuscript document with all these changes highlighted.

MANUSCRIPT ORGANIZATION AND FORMATTING:

To avoid unnecessary delays in the acceptance and publication of your paper, please read the following information carefully. Full guidelines are available on our Instructions for Authors page, <https://www.life-science-alliance.org/authors>

- A number of strains have been attributed to this work in S.Table 2. Please expand the description on the generation of these in the methods section of this study.
- Please label axes for all plots (X-axis in Figures 5C and 5E, Y-axis in Figure S7A, X and Y in Figure S7B, S7C, S8C). When there are multiple related plots in a panel, you can alternatively keep one labelled and in the figure legend, make a note of the related plots that have similar axes titles.
- Please mark panel E in Figure S5.
- Please verify the completeness of Figure S9 and that chromatograms are visible.
- You have the option of using Figure 8 as a Graphical Abstract in which case you would have to remove this figure file and upload it instead as a graphical abstract file, and edit the text accordingly.
- Please upload your main manuscript text as an editable doc file.
- Please upload your main and supplementary figures as single files.
- Please add a Category for your manuscript in our system
- Please add the X and Bluesky handles of your host institute/organization, as well as your own, and/or one of the authors, in our system.
- Please rename "Bibliography" to "References."
- Please be sure that the authorship listing and order is correct

We welcome submissions of potential cover images for the issue of LSA in which your work would appear. If you have high quality images associated with this work, please feel free to email these, with a caption, to the journal office.

LSA encourages authors to provide a 30-60 second video where the study is briefly explained. We will use these videos on social media to promote the published paper and the presenting author (for examples, see

<https://docs.google.com/document/d/1-UWCfbE4pGcDdcgzcmiuJl2XMBJnxKYeqRvLLrLSo8s/edit?usp=sharing>). Corresponding or first-authors are welcome to submit the video. Please submit only one video per manuscript. The video can be emailed to contact@life-science-alliance.org

FINAL FILES:

The following items are required for acceptance.

The license to publish form must be signed before your manuscript can be sent to production. A link to the license to publish form will be available to the corresponding author only. Please take a moment to check your funder requirements.

Thank you for your attention to these final processing requirements. Please revise and format the manuscript and upload materials as soon as you are able.

Thank you for this interesting contribution to the literature. We look forward to publishing your paper in Life Science Alliance.

Sincerely,

Sarita Hebbar, PhD
Scientific Editor
Life Science Alliance
<http://www.lsjournal.org>

Reviewer #1 (Comments to the Authors (Required)):

The authors have carried out a substantial revision, where they addressed all earlier queries satisfactorily.

Reviewer #3 (Comments to the Authors (Required)):

The revised version of the manuscript by Zaremba et al. is clearly improved. The authors have clarified all previously raised points and have also shortened several sections, making the text easier to read. They have addressed all of my concerns, and in my opinion the manuscript can now be published in its current form.

April 27, 2026

RE: Life Science Alliance Manuscript #LSA-2025-03520-TRR

Dr. Melody Cools
Labiris
1 avenue Emile Gryzon
Brussels 1070
Belgium

Dear Dr. Cools,

Thank you for submitting your Research Article entitled "Vsb1, Ypq1 and Ypq2 control dynamic cationic amino acid storage in the yeast vacuole". We apologise for the delay in communicating our decision due to editor availability issues.

It is a pleasure to let you know that your manuscript is now accepted for publication in Life Science Alliance. Congratulations on this interesting work.

Your article will publish open access upon publication under a CC-BY license.

DISTRIBUTION OF MATERIALS:

Again, congratulations on a very nice paper. I hope you found the review process to be constructive and are pleased with how the manuscript was handled editorially. We look forward to future exciting submissions from your lab.

Sincerely,

Sarita Hebbar, PhD
Scientific Editor
Life Science Alliance
<http://www.lsajournal.org>